# Ca²⁺ signaling driving pacemaker activity in submucosal interstitial cells of Cajal in the murine colon

Salah A Baker*, Wesley A Leigh, Guillermo Del Valle, Inigo F De Yturriaga, Sean M Ward, Caroline A Cobine, Bernard T Drumm†, Kenton M Sanders

Department of Physiology and Cell Biology, University of Nevada, Reno, School of Medicine, Reno, United States

**Abstract** Interstitial cells of Cajal (ICC) generate pacemaker activity responsible for phasic contractions in colonic segmentation and peristalsis. ICC along the submucosal border (ICC-SM) contribute to mixing and more complex patterns of colonic motility. We show the complex patterns of Ca²⁺ signaling in ICC-SM and the relationship between ICC-SM Ca²⁺ transients and activation of smooth muscle cells (SMCs) using optogenetic tools. ICC-SM displayed rhythmic firing of Ca²⁺transients ~ 15 cpm and paced adjacent SMCs. The majority of spontaneous activity occurred in regular Ca²⁺ transients clusters (CTCs) that propagated through the network. CTCs were organized and dependent upon Ca²⁺ entry through voltage-dependent Ca²⁺ conductances, L- and T-type Ca²⁺ channels. Removal of Ca²⁺ from the external solution abolished CTCs. Ca²⁺ release mechanisms reduced the duration and amplitude of Ca²⁺ transients but did not block CTCs. These data reveal how colonic pacemaker ICC-SM exhibit complex Ca²⁺-firing patterns and drive smooth muscle activity and overall colonic contractions.

*For correspondence:
sabubaker@med.unr.edu

Present address: † Department of Life & Health Science, Dundalk Institute of Technology, Dundalk, Ireland

Competing interests: The authors declare that no competing interests exist.

## Introduction

Interstitial cells of Cajal (ICC) serve several important functions in the gastrointestinal tract, including generation of pacemaker activity (*Langton et al., 1989*; *Ward et al., 1994*; *Huizinga et al., 1995*), neurotransduction (*Burns et al., 1996*; *Ward et al., 2000a*; *Sanders et al., 2010*), and responses to stretch (*Won et al., 2005*). Electrical activity in ICC is transmitted to other cells in the *tunica muscularis* via gap junctions (*Komuro, 2006*). Smooth muscle cells (SMCs), ICC and another class of platelet-derived growth factor receptor alpha (PDGFRα)-positive interstitial cells are linked together in a coupled network known as the Smooth muscle cells, ICC and PDGFRα⁺ cells (SIP) syncytium (*Sanders et al., 2012*). The pacemaker function of ICC was deduced from morphological studies (*Faussone Pellegrini et al., 1977*; *Thuneberg, 1982*), dissection of pacemaker regions experiments (*Smith et al., 1987a*; *Hara et al., 1986*; *Serio et al., 1991*), studies on isolated ICC and SMCs (*Langton et al., 1989*; *Sanders et al., 2014a*; *Zhu et al., 2009*), studies of muscles from animals with loss-of-function mutations in c-Kit signaling (*Ward et al., 1994*; *Ward et al., 1995*; *Huizinga et al., 1995*) and simultaneous impalements of ICC and SMCs (*Cousins et al., 2003*). While these experiments were strongly indicative of the obligatory role of ICC as pacemakers in GI smooth muscles and essential role in normal electrical and contractile patterns, no experiments to date have measured pacemaker activity in ICC and responses of SMCs in terms of Ca²⁺ signaling and contraction simultaneously.

The anatomy and distribution of ICC varies from place to place throughout the GI tract. Some areas have only an intramuscular type of ICC (ICC-IM) that are closely aligned with and transduce inputs from excitatory and inhibitory enteric motor neurons (*Burns et al., 1996*; *Beckett et al., 2004*; *Lies et al., 2014*; *Sanders et al., 2014b*). Other regions contain ICC-IM and pacemaker types

of ICC, that exist as a network in the myenteric plexus region of most areas of the gut (ICC-MY) (*Rumessen and Thuneberg, 1982*; *Faussone-Pellegrini, 1992*; *Komuro et al., 1996*; *Burns et al., 1997*; *Komuro, 1999*). The colon is more complex in that there are at least four types of ICC, distinguished by their anatomical locations and functions (*Ishikawa and Komuro, 1996*; *Vanderwinden et al., 2000*; *Vanderwinden et al., 1996*; *Rumessen et al., 2013*). One class of colonic ICC lies along the submucosal surface of the circular muscle (CM) layer (ICC-SM). These cells are known to provide pacemaker activity in colonic muscles, and their activity is integrated with a second frequency of pacemaker activity generated by ICC-MY (*Smith et al., 1987a*; *Smith et al., 1987b*; *Durdle et al., 1983*; *Huizinga et al., 1988*; *Yoneda et al., 2002*; *Plujà et al., 2001*). Pacemaker activities generated by ICC-SM and ICC-MY cause depolarization of SMCs, generation of $Ca^{2+}$ action potentials, and excitation-contraction coupling (*Yoneda et al., 2004*).

ICC-SM generate slow waves in the canine colon (*Smith et al., 1987a*; *Berezin et al., 1988*). These are large amplitude and long-duration events that produce phasic contractions (*Keef et al., 1992*). The integrity of the ICC-SM network is required for regenerative propagation of slow waves, and disruption of the network causes passive decay of slow waves within a few millimeters (*Sanders et al., 1990*). Electrical coupling of ICC-SM in a network is an important feature allowing the pacemaker activity to coordinate the electrical activation of SMCs. ICC-SM in proximal colons of rodents also display pacemaker function; however, the frequency of the slow waves is higher (10–22 $min^{-1}$, mean 14.8. $min^{-1}$) (*Yoneda et al., 2002*). Slow waves, in this region of the GI tract, consist of a rapid upstroke phase, 148 $mVs^{-1}$, that settles to a plateau phase lasting approximately 2 s. The slow waves are coupled to low-amplitude CM contractions (*Yoneda et al., 2004*). Colonic slow waves have been reported to depend upon both $Ca^{2+}$ entry and intracellular $Ca^{2+}$ release mechanisms; however, $Ca^{2+}$ signaling in colonic pacemaker cells and the coupling of $Ca^{2+}$ events to the electrical responses were not clarified.

Previous studies have shown that all classes of ICC in the GI tract express $Ca^{2+}$-activated $Cl^-$ channels encoded by *Ano1* (*Gomez-Pinilla et al., 2009*; *Hwang et al., 2009*). This conductance is required for slow wave activity (*Hwang et al., 2009*), and therefore $Ca^{2+}$ dynamics in ICC are of fundamental importance in understanding pacemaker activity and electrical and mechanical rhythmicity in GI muscles. In the present study, we tested the hypothesis that $Ca^{2+}$ transients in ICC-SM are linked to mechanical activation of the CM and that propagation of activity in ICC-SM is related to and controlled by $Ca^{2+}$ entry via voltage-dependent $Ca^{2+}$ conductances. Experiments were performed on tissues containing ICC-SM taken from mice with cell-specific expression of GCaMP6f in ICC, and changes in intracellular $Ca^{2+}$ were monitored by confocal microscopy and digital video imaging.

## Results

### ICC-SM distribution within the submucosal plexus

We optimized a preparation in which the submucosal layer was separated from the *tunica muscularis*. We found that ICC-SM were adherent to the submucosal tissues, so this preparation allowed very clear high resolution of $Ca^{2+}$ transients in ICC-SM in the complete absence of motion artifacts due to muscle contractions. We confirmed the presence and maintenance of ICC-SM networks, which occur in intact muscles, in these preparations.

Kit immunoreactivity revealed a dense network of Kit-positive cells in the submucosal layer of the proximal colon (*Figure 1A*). The network consisted of ICC-SM interconnected with branching processes (*Figure 1B*). The average density of cell bodies was 312 ± 33 cells $mm^{-2}$ (*n* = 6), and the average minimum separation between cell bodies was 49.3 ± 2.8 μm (*Figure 1A&B*; *n* = 6). Most of the ICC-SM network appeared to be adherent to the submucosal layer of the proximal colon. Isolation of the submucosa by sharp dissection showed that few ICC-SM (1.6 ± 2.1 cells $mm^{-2}$; *Figure 1C*; *n* = 6) remained adherent to the muscularis in the uppermost region (1–2 μm) of the proximal colon.

Colonic muscles from $Kit^{+/copGFP}$ mice expressing the copGFP exclusively in ICC were also used to confirm the distribution of ICC-SM. copGFP-positive cells in the submucosal region were present at an average density of 284 ± 27 cells $mm^{-2}$ and the average minimum separation between cell bodies was 52.7 ± 2.9 μm (*Figure 1D and E*; *n* = 6). ICC-SM were not resolved at the surface of the muscle layer after removing the submucosal layer (*Figure 1F*; *n* = 6). Colonic muscles expressing

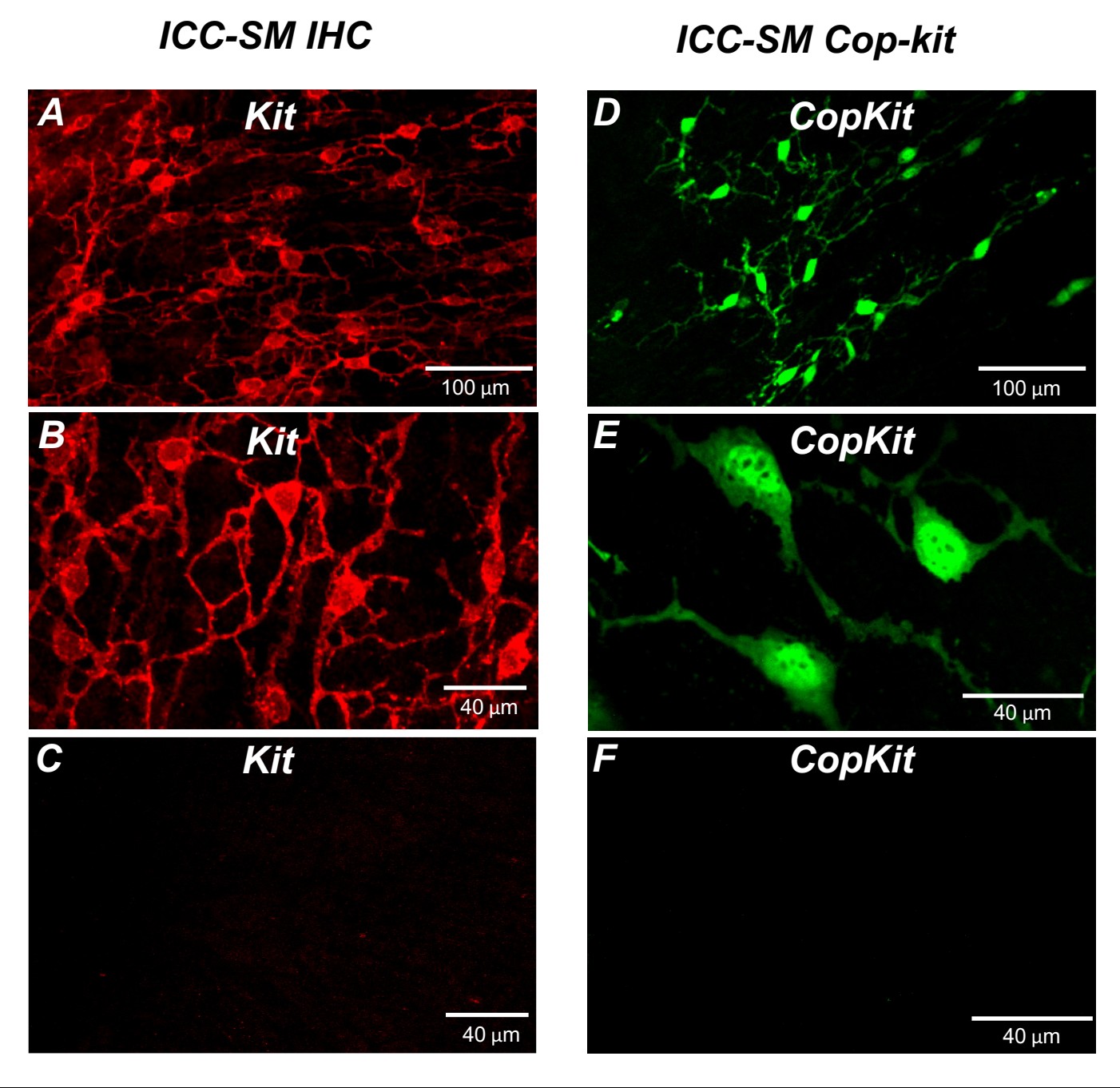

**Figure 1.** Distribution of Kit$^+$ submucosal interstitial cells of Cajal (ICC-SM) in the murine colon. (A and B) Images of Kit$^+$ (ICC-SM) in isolated submucosal layer of wild-type animals at ×20 and ×40 magnifications ($n$ = 6). Scale bars are 100 μm and 40 μm respectively. (C) Absence of ICC-SM on the submucosal surface of the *tunica muscularis* of the proximal colon after removing the submucosa. ICC-SM networks are intact in preparations of submucosal tissues removed from the muscle. (D and E) ICC-SM were present in Kit$^{+/copGFP}$ mice at ×20 and×60 magnifications ($n$ = 6). Scale bars are 100 μm and 40 μm respectively. (F) Absence of Kit$^+$ (ICC-SM) on the submucosal surface of the *tunica muscularis* after removing the submucosa from Kit$^{+/copGFP}$ proximal colon muscles ($n$ = 6). Scale bars are 40 μm in both C and F. All image parameters were analyzed using Image J software.

GCaMP6f exclusively in ICC were used to monitor Ca$^{2+}$ signaling in ICC-SM. GCaMP6f-positive cells in the submucosal region were present at an average density of 291 ± 36 cells mm$^{-2}$ ($n$ = 10), and the average minimum separation between cell bodies was 50.6 ± 3.8 μm. Representative images of

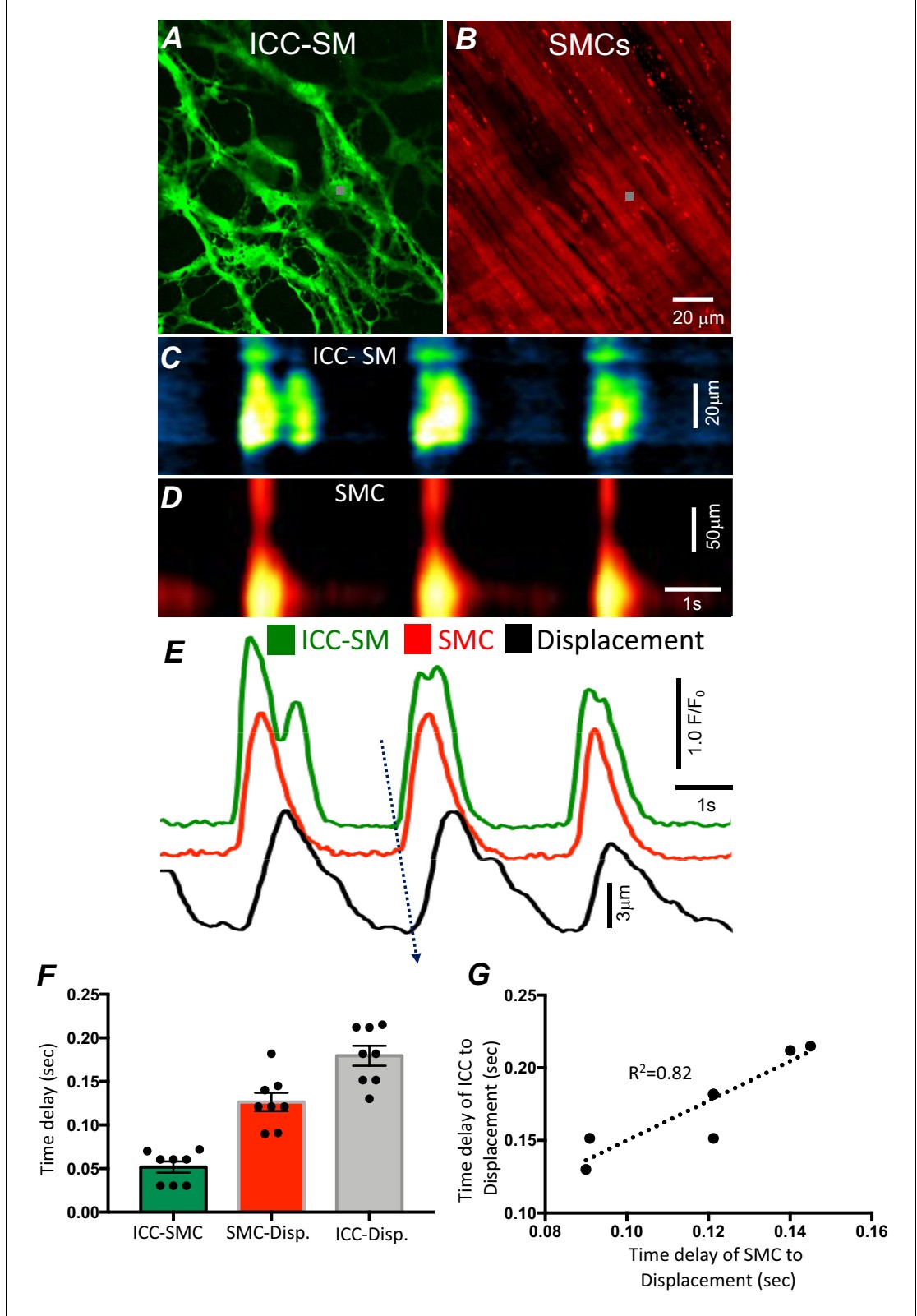

**Figure 2.** Temporal sequence of $Ca^{2+}$ transients firing in submucosal interstitial cells of Cajal (ICC-SM) and smooth muscle cells (SMCs). Representative dual-color imaging of ICC-SM (green; **A**) and SMCs (red; **B**) $Ca^{2+}$ transients were recorded simultaneously from proximal colonic muscles of *Kit-iCre-GCaMP6f/Acta2-RCaMP1.07* mice (see Materials and methods for details). **C and D** Spatiotemporal maps (STMaps) of $Ca^{2+}$ signals in ICC-SM and SMCs during a recording period (gray boxes in **A and B** denote cell locations; see *Video 1*). The STMaps show coordinated firing of $Ca^{2+}$ transients in

*Figure 2 continued on next page*

*Figure 2 continued*

both types of cells. Ca$^{2+}$ transient traces are plotted in *E* (ICC-SM-green, SMCs-red). Tissue displacement was also monitored and plotted (black trace; *E*). The black dotted arrow depicts the sequence of activation firing of Ca$^{2+}$ transients in ICC-SM, to activation of Ca$^{2+}$ events in SMCs, and tissue displacement. (*F*) A comparison of latencies (ms) from the start of the initiation of Ca$^{2+}$ transients in ICC-SM to SMC activation and tissue displacement (*n* = 8). (*G*) Correlation analysis of latencies between ICC-SM and SMC Ca$^{2+}$ transients and tissue displacement (R$^2$ = 0.82). All data graphed as mean ± SEM.

the GCaMP6f-expressing cells in ICC-SM are shown below in the Ca$^{2+}$-imaging experiments (*Figure 2A*; *Figure 3A* and Figure 6A).

## Pacemaker activity of ICC-SM drives responses in SMCs

We developed a new mouse strain (*Kit-iCre-GCaMP6f/Acta2-RCaMP1.07*) that allowed simultaneous optical measurements of Ca$^{2+}$ transients in ICC-SM and SMCs and tissue displacement (i.e. an optical means of monitoring muscle contractions). The dual-color mouse allowed simultaneous imaging of the two optogenetic Ca$^{2+}$ sensors with different fluorescence characteristics (ensuring minimal spectral overlap). The sensors were expressed in a cell-specific manner in ICC and SMCs to characterize the coordination of signaling between ICC and SMCs (*Figure 2* and *Video 1*). Ca$^{2+}$ imaging of colonic muscles with attached submucosa showed a strong correlation between Ca$^{2+}$ transients in the ICC-SM network and SMCs (*Figure 2A–E*; *n* = 8). Ca$^{2+}$ transients recordings were obtained from ICC-SM and SMC at the exact coordinates in the pixel matched images (indicated by the gray box; *Figure 2A&B*) The sequence of activation began in ICC-SM and spread to SMCs with a latency of 56 ± 14 ms (*Figure 2E&F*; *n* = 8). Muscle displacement, an indicator of muscle contraction, was also measured and displayed latencies of 120 ± 17 ms between the rise of Ca$^{2+}$ in SMCs and resolvable displacement (*Figure 2E&F*; *n* = 8) and 180 ± 15 ms between activation of Ca$^{2+}$ transients in ICC-SM and contractile displacement (*Figure 2E&F*; *n* = 8). In a few instances, the time delay between ICC-SM and SMCs was not readily resolved. This could have resulted from a shift in the site of dominant pacemaker activity to a region outside the field of view (FOV). There was close correlation between the temporal latencies of ICC-SM and SMCs to tissue displacement (*Figure 2G*; R$^2$ = 0.82), indicating the pacemaker nature of ICC-SM, the coupling between ICC-SM and SMCs and the management of Ca$^{2+}$ signaling and contractions in SMCs by the pacemaker activity of ICC-SM.

## Global Ca$^{2+}$-firing patterns in ICC-SM

Movement generated by muscle contractions is always an issue when imaging cells in muscle preparations in situ. Therefore, we used preparations of ICC-SM adherent to submucosa that were separated from muscle strips in most experiments. At low-magnification (×10), rhythmic Ca$^{2+}$ waves occurred and spread through ICC-SM networks in isolated submucosal preparations. The Ca$^{2+}$ waves occurred at 8–22 cycle min$^{-1}$ (*Figure 3A–G*) and averaged 14.9 ± 1.9 cycle min$^{-1}$ (*Figure 3H*; *n* = 15; c = 120). Similar behavior occurred at similar frequencies (i.e. 16.2 ± 1.4 cycle min$^{-1}$; *n* = 6) in ICC-SM attached to muscle tissues. Ca$^{2+}$ waves propagated through the isolated ICC-SM networks with velocities of 219 ± 19 mm/s. Ca$^{2+}$ transients imaged at ×10 appeared to be global and had average durations of 2.1 ± 0.4 s (*Figure 3I*; *n* = 15; c = 120) and amplitudes of 1.2 ± 0.3 ΔF/F$_0$ (*Figure 3J*; *n* = 15; c = 120). The spatial spread of Ca$^{2+}$ transients was 36.8 ± 0.4 μm (*Figure 3K*; *n* = 15; c = 120). The average area occupied by Ca$^{2+}$ transients in the spatiotemporal maps was 51.6 ± 1.6 μm*s (*Figure 3L*; *n* = 15; c = 120).

Cell-to-cell propagation of Ca$^{2+}$ transients in ICC-SM is shown in *Figure 3* and *Figure 4*. Firing of global Ca$^{2+}$ transients appeared to be sequential in nature (*Figure 4A–G*; *n* = 10; c = 60), as there was a strong correlation between the occurrence of global Ca$^{2+}$ transients in multiple ICC-SM cells (*Figure 4B*; *n* = 10; c = 60). Overlays of STMaps of Ca$^{2+}$ activity in adjacent cells running parallel to each other (*Figure 4F&H*) showed that the Ca$^{2+}$ transients overlapped (65.4% of total Ca$^{2+}$transients overlapped in the FOV during a 30-s recording period) (*Figure 4G*). Comparison between intervals of Ca$^{2+}$ firing of multiple ICC-SM showed strong sequential firing as each cell demonstrated very close temporal firing intervals (*Figure 4H&I*; *n* = 6; c = 18). These results suggest that Ca$^{2+}$ firing in ICC-SM networks are entrained and propagate cell-to-cell.

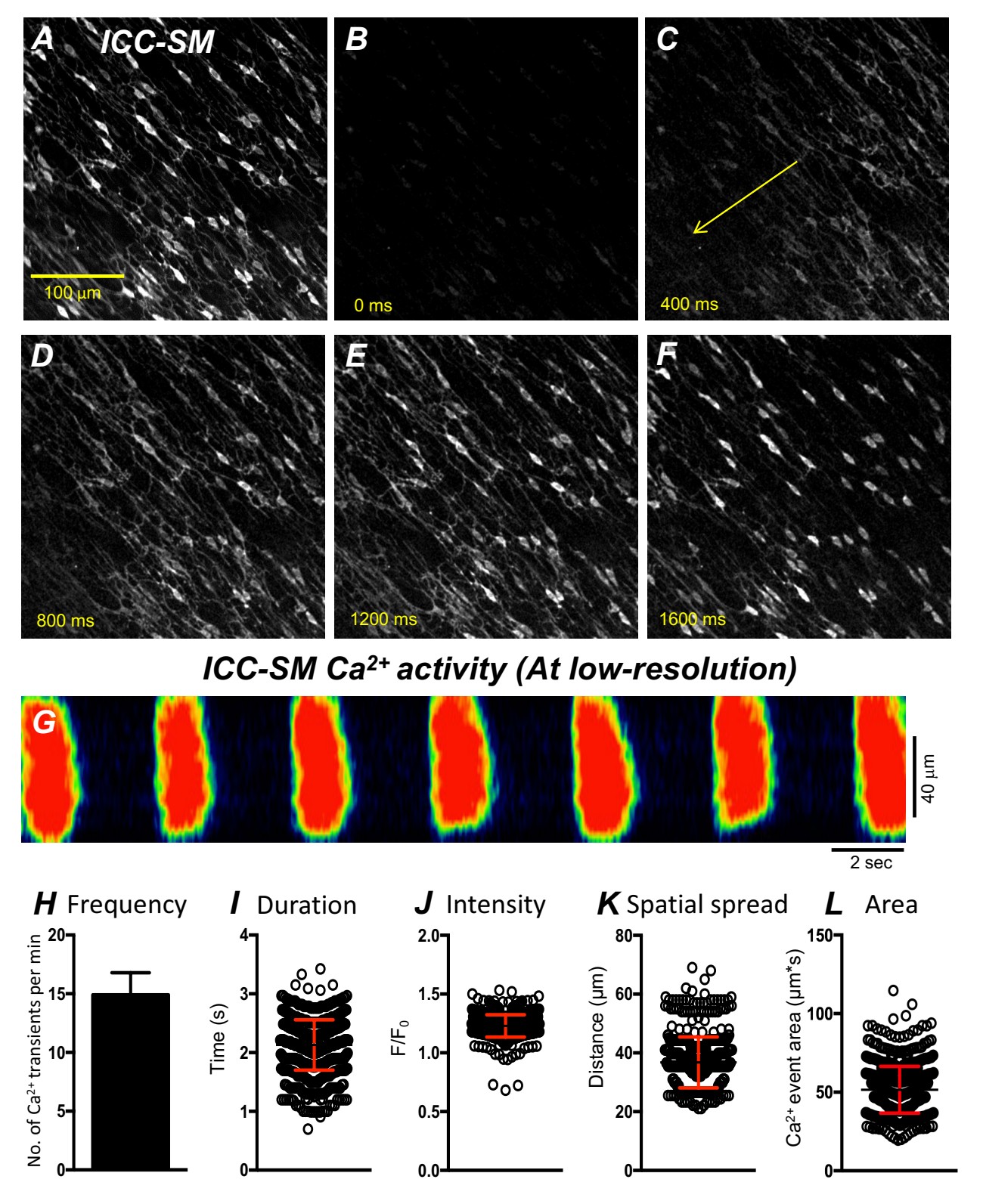

**Figure 3.** Propagating Ca$^{2+}$ waves in submucosal interstitial cells of Cajal (ICC-SM) network. (**A**) An image of ICC-SM network from the proximal colon of a Kit-iCre-GCaMP6f mouse visualized at 10x (low resolution) magnification. Scale bar is 100 µm in **A** and pertains to all images in **B-F**. **A-F** Representative montage of the propagation of a Ca$^{2+}$ wave throughout the ICC-SM network. The yellow arrow in panel C indicates the direction of Ca$^{2+}$ transient propagation. (**G**) Spatiotemporal map (STMap) of Ca$^{2+}$ signal in single ICC from the movie in panel **A** showing rhythmic firing of Ca$^{2+}$

*Figure 3 continued on next page*

*Figure 3 continued*

waves. $Ca^{2+}$ activity is color coded with warm areas (red, orange) representing bright areas of $Ca^{2+}$ fluorescence and cold colors (blue, black) representing dim areas of $Ca^{2+}$ fluorescence. Summary data of multiple ICC-SM $Ca^{2+}$-firing parameters ($n = 15$): frequency **H**, duration **I**, intensity **J**, spatial spread **K** and area of $Ca^{2+}$ transients **L**. All data graphed as mean ± SEM.

## ICC-SM $Ca^{2+}$ signals composed of multiple $Ca^{2+}$-firing sites

Low-power imaging suggested global changes in cell $Ca^{2+}$, however, imaging of ICC-SM at higher power (×60–×100) allowed a more detailed visualization of the subcellular nature of the $Ca^{2+}$ transients and revealed complex firing patterns. Imaging of ICC-SM networks with high spatial resolution showed that subcellular $Ca^{2+}$ transients originated from distinct firing sites (*Figure 5*; $n = 25$). STMaps constructed from $Ca^{2+}$ transients in single ICC-SM during 30 s of imaging (*Figure 5A&B*) identified the positions of all frequent firing sites within ICC-SM. Activity plots of individual firing sites showed they can fire once or multiple times during single $Ca^{2+}$ waves (*Figure 5B&C*). The number of firing sites in a single ICC-SM ranged from 5 to 12 sites with an overall average of 8.2 ± 2 sites/cell (*Figure 5D*; $n = 25$).

We employed $Ca^{2+}$ particle analysis (*Drumm et al., 2017*) to identify and quantify $Ca^{2+}$-firing sites in ICC-SM. $Ca^{2+}$-firing sites were distinguished based on their spatial and temporal characteristics (*Figure 6A–D* and *Video 2*). The firing sites were color coded, as shown in the example in *Figure 6D*, to visualize and quantify all active sites firing $Ca^{2+}$ transients. The activities of firing sites were plotted as occurrence maps (*Figure 6E*). Occurrence maps of all firing sites in a region of ICC-SM network demonstrated that the global $Ca^{2+}$ waves resolved at low power resulted from summation of localized $Ca^{2+}$ transients from a multitude of firing sites (*Figure 6D–J*). $Ca^{2+}$ transients were clustered temporally as $Ca^{2+}$ waves swept through ICC-SM networks (*Figure 6E&J*). Also apparent from the occurrence maps was that the firing sequence of sites changed from $Ca^{2+}$ wave to $Ca^{2+}$ wave. Not all firing sites discharged $Ca^{2+}$ transients during each wave cycle, and some sites fired more than once (*Figure 6E&J*). From particle analysis the average particle area/frame of $Ca^{2+}$ transients averaged 3.2 ± 0.4 μm$^2$ (*Figure 6H*; $n = 25$) and particle count/frame averaged 0.27 ± 0.1 (*Figure 6I*; $n = 25$).

Particle analysis also showed that $Ca^{2+}$-firing sites were most active during the first ~256 ms of a $Ca^{2+}$ wave, and activity decayed with time. This point was further illustrated by distribution plots showing the average percentage of firing sites discharging at various times during $Ca^{2+}$ waves (*Figure 6K*; $n = 25$). Initially high firing and decay as a function of time suggest that $Ca^{2+}$ entry mechanisms may be important for: (i) initiating $Ca^{2+}$ transients and (ii) organizing the occurrence of $Ca^{2+}$ transients into clusters. It is also possible that $Ca^{2+}$ stores, loading during the diastolic period between $Ca^{2+}$ waves, are more excitable at the onset of each $Ca^{2+}$ wave.

## $Ca^{2+}$ influx mechanisms are required for initiation of clustered $Ca^{2+}$ transients in ICC-SM

The effects of reducing extracellular $Ca^{2+}$ ($[Ca^{2+}]_o$) on $Ca^{2+}$ transients was investigated to evaluate the importance of $Ca^{2+}$ influx for the

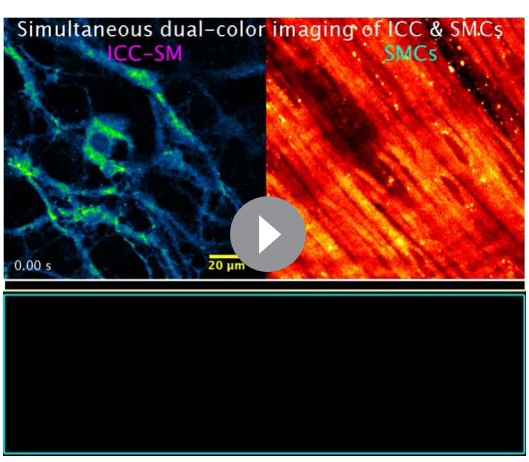

**Video 1.** Simultaneous dual-color imaging of submucosal interstitial cells of Cajal (ICC-SM) and smooth muscle cells (SMCs) in the colon. A video of propagating $Ca^{2+}$ waves through an ICC-SM network in the proximal colon of *Kit-iCre-GCaMP6f/Acta2-RCaMP1.07* strain imaged with a ×20 objective. simultaneous imaging of two optogenetic $Ca^{2+}$ sensors: *GCaMP6f* in ICC-SM (left field of view [FOV]; green) and *RCaMP1.07* in SMCs (right FOV; red) with different fluorescence characteristics (ensuring minimal spectral overlap). The signal coordination between ICC and SMCs showing the correlation between $Ca^{2+}$ transients in the ICC-SM network and activation of SMCs adjacent to ICC-SM. ICC-SM transients (green trace) preceded $Ca^{2+}$ signals in SMCs (red trace). The scale bar (yellow) is 25 μm.

https://elifesciences.org/articles/64099#video1

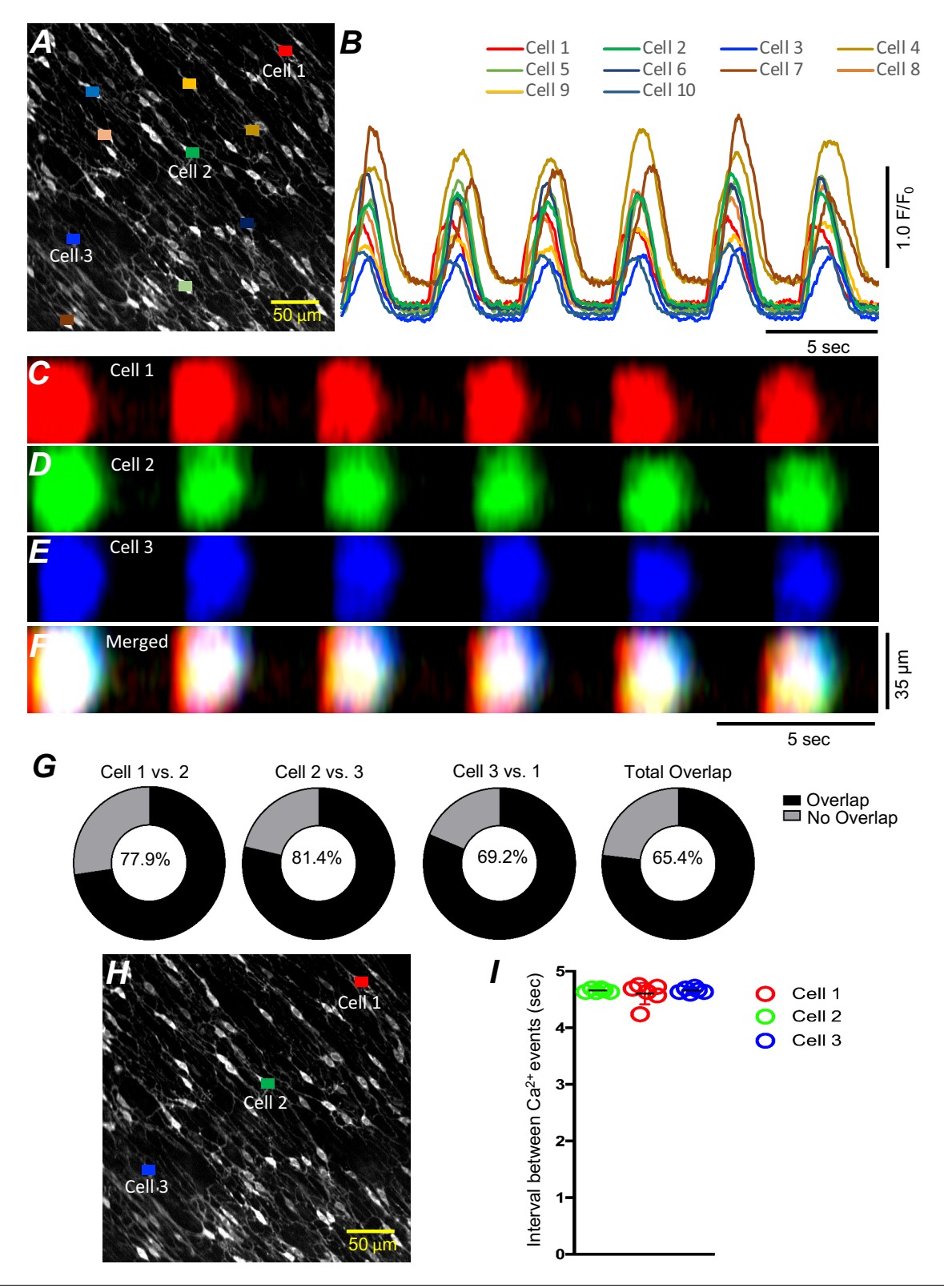

**Figure 4.** Submucosal interstitial cells of Cajal (ICC-SM) Ca$^{2+}$ signaling activity is entrained. (**A**) Raw image of multiple colonic ICC-SM in a field of view (FOV). Ten cells were color coded, and their Ca$^{2+}$ fluorescence activity traces were plotted in **B**. Three ICC-SM (coded as red, green and blue regions of interest; ROIs in **A**) were selected and STMaps of the average Ca$^{2+}$ fluorescence intensity across the diameter of the cell during a 30-s recording were constructed **C–E**. STMaps from each cell were color coded to correspond to the red, green, and blue cells and merged into a summed

*Figure 4 continued on next page*

Figure 4 continued

STMap in **F**. Percentage of fluorescence area overlap of intracellular $Ca^{2+}$ transients between ICC-SM cells is plotted in **G** and cell location is identified in **H** ($n = 10$). The durations of $Ca^{2+}$ waves were such that there was a significant overlap of the $Ca^{2+}$ events in individual cells across the FOVs at this magnification. Thus, each cell within the FOVs demonstrated similar temporal firing intervals (calculated as peak to peak intervals, $n = 6$) as shown in **I**.

patterning of $Ca^{2+}$ signaling in ICC-SM. Removal of $[Ca^{2+}]_o$ abolished $Ca^{2+}$ transients in ICC-SM within 10 min (*Figure 7E–L*; $n = 6$). Stepwise reduction in $[Ca^{2+}]_o$ (from 2.5 mM to 0 mM) showed that $Ca^{2+}$ transients decreased in a concentration-dependent manner (*Figure 7*; $n = 6$). ICC-SM $Ca^{2+}$ PTCL area and count of firing sites were reduced in response to lowering $[Ca^{2+}]_o$ (*Figure 7F–J*; $n = 6$). Reducing $[Ca^{2+}]_o$ from 2.5 mM (control conditions) to 2 mM caused a reduction in the average PTCL area by $19.6 \pm 4.5\%$ (*Figure 7K*; $n = 6$) and PTCL count by $25.4 \pm 2.9\%$ (*Figure 7L*; $n = 6$) with no significant change in the duration of PTCLs (*Figure 7M*; $n = 7$). Further reduction of $[Ca^{2+}]_o$ to 1 mM reduced $Ca^{2+}$ transient parameters by $47.2 \pm 3.5\%$ PTCL area (*Figure 7K*; $n = 6$) and $45.1 \pm 4.4\%$ PTCL count (*Figure 7L*; $n = 6$) and showed a significant change in the duration of PTCLs (*Figure 7M*; $n = 7$). Lowering $[Ca^{2+}]_o$ to 0.5 mM also reduced $Ca^{2+}$ PTCL area by $73.9 \pm 3\%$ (*Figure 7K*; $n = 6$) and PTCL count by $73.2 \pm 2\%$ (*Figure 7L*; $n = 6$) with no significant change in the duration of PTCLs (*Figure 7M*; $n = 7$). Removal of $Ca^{2+}$ from the extracellular solution ($Ca^{2+}$-free KRB solution containing 0.5 mM EGTA) abolished $Ca^{2+}$ signals 8–10 min after solution replacement. The $Ca^{2+}$ transient PTCL area was reduced by $99.4 \pm 0.6\%$ (*Figure 7K*; $n = 6$) and PTCL count by $99.3 \pm 0.7\%$ (*Figure 7L*; $n = 6$) with no significant change in the duration of PTCLs (*Figure 7M*; $n = 7$). We also noted that the highly organized CTCs occurring in the presence of 2.5 mM $[Ca^{2+}]_o$ became less organized as $[Ca^{2+}]_o$ was reduced (e.g. compare the tight clusters in *Figure 7A* with the more diffuse clusters in *Figure 7C*). These experiments highlight the importance of $Ca^{2+}$ entry mechanisms in $Ca^{2+}$ signaling within ICC-SM.

## Molecular expression of $Ca^{2+}$ entry channels in ICC-SM

The apparent dependence on the $Ca^{2+}$ gradient to maintain pacemaker function in ICC-SM suggests that $Ca^{2+}$ entry mechanisms are critical for initiation and organization of CTCs. Therefore, we examined expression of several $Ca^{2+}$ channels that might be responsible for $Ca^{2+}$ entry in ICC-SM (*Figure 8*). After isolation of the submucosal layer from the proximal colon of Kit[+/copGFP] mice and subsequent cell dispersion, we sorted copGFP-positive ICC-SM with fluorescence activated cell-sorting (FACS) and evaluated the expression of voltage-dependent and voltage-independent $Ca^{2+}$ channels by qPCR (*Figure 8A&B*). First, we confirmed the purity of sorted ICC-SM with cell-specific markers (*Figure 8A*). Kit receptors and ANO1 channels are signatures of ICC throughout the GI tract and enrichment of *Kit* and *Ano1* expression was observed in sorted ICC-SM compared to unsorted cells (*Kit* expression was $0.21 \pm 0.014$; *Ano1* expression was $0.14 \pm 0.02$ relative to *Gapdh*). The expression levels of *Myh11* an SMC marker and *Uch11* a pan neuronal marker encoding PGP9.5 were minimal (*Myh11* expression was $0.03 \pm 0.002$; Uch*11* expression was $0.002 \pm 0.0001$ relative to *Gapdh*), confirming the purity of ICC-SM sorted by FACS.

ICC-SM expressed L-type voltage-dependent $Ca^{2+}$ channels encoded by *Cacna1c* and *Cacna1d* abundantly ($Ca_V 1.2$ and $Ca_V 1.3$ channels, respectively). *Cacna1c* showed a $0.012 \pm 0.0005$ and *Cacna1d* showed $0.066 \pm 0.008$ relative to *Gapdh* (*Figure 8B*; $n = 4$). ICC-SM also expressed *Cacna1h* ($Ca_V 3.2$) and to a lesser extent *Cacna1g* ($Ca_V 3.1$), both T-type voltage-dependent $Ca^{2+}$ channels (*Figure 8B*; $n = 4$). *Cacna1h* expression was abundant in ICC-SM ($0.07 \pm 0.003$ relative to *Gapdh*). *Cacna1g* expression was less than *Cacna1h* $0.0014 \pm 0.0001$ relative to *Gapdh* (*Figure 8B*; $n = 4$).

Maintenance and refilling of cellular $Ca^{2+}$ stores may also be important for mediation and shaping of $Ca^{2+}$ signals in ICC-SM. Store-operated $Ca^{2+}$ entry (SOCE) has been established as a mechanism for filling stores upon depletion (*Putney, 2018*; *Gibson et al., 1998*). Contributions from SOCE via STIM and Orai interactions are essential for maintenance of $Ca^{2+}$ stores in other types of ICC (*Zheng et al., 2018*; *Youm et al., 2019*; *Drumm et al., 2020a*; *Drumm et al., 2020b*). Colonic ICC-SM showed enrichment in *Orai1* and *Orai2* but *Orai3* was not resolved (*Figure 8B*; $n = 4$). *Orai1 expression* was $0.02 \pm 0.001$ relative to *Gapdh* and *Orai2* was $0.074 \pm 0.008$ relative to *Gapdh* (*Figure 8B*; $n = 4$).

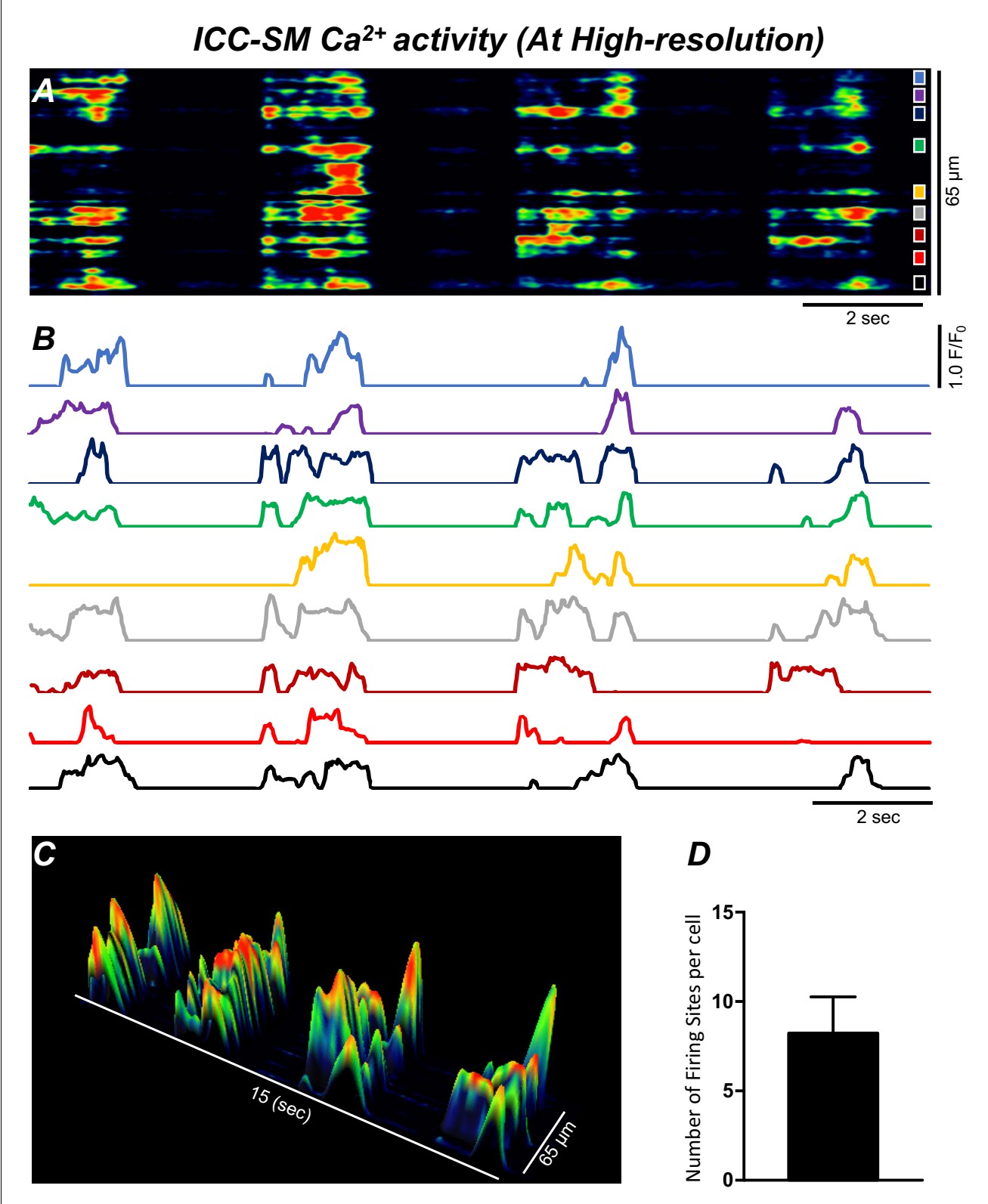

**Figure 5.** Ca²⁺ transients in submucosal interstitial cells of Cajal (ICC-SM) arise from multiple firing sites. (**A**) Spatiotemporal map (STMap) of Ca²⁺-firing sites from a single ICC-SM during four consecutive firing cycles. Ca²⁺ activity is color coded with warm areas (red, orange) representing bright areas of Ca²⁺ fluorescence and cold colors (blue, black) representing dim areas of Ca²⁺ fluorescence. Nine distinct firing sites were detected in this cell and are marked as color squares on the right of the STMap. (**B**) Traces of the Ca²⁺ transients at each of the nine Ca²⁺-firing sites shown on the STMap

*Figure 5 continued on next page*

*Figure 5 continued*
in panel A. (C) 3-D surface plots showing the $Ca^{2+}$ activity at the $Ca^{2+}$-firing sites shown on the STMap in panel **A** over four consecutive $Ca^{2+}$-firing cycles. (D) Average number of the firing sites per cell ($n = 25$). Data graphed as mean ± SEM.

## L-type $Ca^{2+}$ channels are important for organization of $Ca^{2+}$ transients in ICC-SM

L-type $Ca^{2+}$ channels ($Ca_V$ 1.3, $Ca_V$ 1.2) were expressed in ICC-SM, and $Ca^{2+}$ transients showed dependence on the $Ca^{2+}$ gradient. Previous studies have reported that L-type channel antagonists inhibit slow waves in the colon (*Yoneda et al., 2002*; *Keef et al., 2002*). Therefore, we evaluated the contributions of $Ca^{2+}$ entry via L-type channels to $Ca^{2+}$ transients in ICC-SM. Nicardipine (1 µM) abolished $Ca^{2+}$ transients in ICC-SM (*Figure 9A&B*; $n = 8$). Firing site occurrence maps (*Figure 9C&D*; $n = 8$) describe the inhibitory effects of nicardipine on $Ca^{2+}$ transients. $Ca^{2+}$ PTCL area was reduced to $10.5 \pm 4.7\%$ (*Figure 9E–G*; $n = 8$) and PTCL count was reduced to $12.3 \pm 4.8\%$ (*Figure 9H*; $n = 8$). The number of firing sites also decreased to $8.4 \pm 3\%$ in the presence of nicardipine (*Figure 9I*; $n = 8$). Isradipine inhibits $Ca_V$ 1.2 and $Ca_V$ 1.3 equally (*Rüegg and Hof, 1990*; *Koschak et al., 2001*). Isradipine (1 µM) also reduced $Ca^{2+}$ transients in ICC-SM (*Figure 9—figure supplement 1A&B*; $n = 7$). ICC-SM $Ca^{2+}$-firing was reduced in the presence of isradipine as shown in the firing sites occurrence maps (*Figure 9—figure supplement 1C&D*). PTCL area was reduced to $18 \pm 5\%$ (*Figure 9—figure supplement 1E–G*; $n = 7$), and PTCL count was reduced to $19.5 \pm 6\%$ (*Figure 9—figure supplement 1H*; $n = 7$). The number of firing sites was inhibited by isradipine to $21.5 \pm 6\%$ (*Figure 9—figure supplement 1I*; $n = 7$). These data show that ICC-SM $Ca^{2+}$ transients depend upon $Ca^{2+}$ influx via L-type $Ca^{2+}$ channels, and since isradipine had nearly the same or a lesser effect on $Ca^{2+}$ transients than nicardipine, $Ca_V$ 1.2 appear dominant as the $Ca^{2+}$ entry mechanism.

## T-type $Ca^{2+}$ channels contribute to $Ca^{2+}$ transients in ICC-SM

T-type $Ca^{2+}$ channels have been identified as the dominant voltage-dependent $Ca^{2+}$ conductance responsible for $Ca^{2+}$ entry and propagation of slow waves in ICC-MY of the small intestine (*Kito et al., 2005*; *Zheng et al., 2014*; ). T-type channel antagonists ($Ni^{2+}$ and mibefradil) reduced the rate-of-rise of the upstroke component of the slow waves and higher concentrations attenuated slow wave activity (*Kito et al., 2005*; *Hotta et al., 2007*; *Yoneda et al., 2003*). As reported above, ICC-SM express T-type channels *Cacna1h* and *Cacna1g*. Therefore, the role of T-type $Ca^{2+}$ channels in modulating $Ca^{2+}$signaling in ICC-SM was evaluated with specific T-type channel antagonists, NNC 55–0396 (10 µM), TTA-A2 (10 µM) and Z-944 (1 µM). NNC 55–0396 reduced $Ca^{2+}$ transient firing (*Figure 10A&B*), $Ca^{2+}$ transients firing sites occurrence maps (*Figure 10C&D*) and firing sites. PTCL area and PTCL count traces show a reduction in $Ca^{2+}$ transient firing (*Figure 10E&F*). PTCL area was reduced to $31.7 \pm 3.3\%$ (*Figure 10E–G*; $n = 9$) and PTCL count was reduced to $35.6 \pm 4.5\%$ (*Figure 10H*; $n = 9$). The number of firing sites was reduced by NNC 55–0396 to $37.6 \pm 4.3\%$ (*Figure 10I*; $n = 9$). The duration of PTCLs was also reduced by NNC 55–0396 to $73 \pm 5.2\%$ (*Figure 10J*; $n = 9$). TTA-A2 showed similar inhibitory effects on ICC-SM $Ca^{2+}$ transients (*Figure 10—figure supplement 1*). PTCL area was reduced to $42 \pm 5.8\%$ (*Figure 10—figure supplement 1E*; $n = 7$) and PTCL count was reduced to $44.4 \pm 6.2\%$ (*Figure 10—figure supplement 1F*; $n = 7$) The number of firing sites was also reduced by TTA-A2 to $43.4 \pm 6\%$ (*Figure 10—figure supplement 1G*; $n = 7$). Z-944 significantly reduced ICC-SM $Ca^{2+}$ transients but was somewhat less effective than NNC 55–0396 or TTA-A2. PTCL area was reduced to $56 \pm 10\%$ (*Figure 10—figure supplement 1E*; $n = 5$), and PTCL count was reduced to $53 \pm 12\%$ (*Figure 10—figure supplement 1F*; $n = 5$) The number of firing sites was also reduced by Z-944 to $53.4 \pm 11\%$ (*Figure 10—figure supplement 1G*; $n = 5$). The data suggest that T-type $Ca^{2+}$ channels also contribute to the initiation and organization of $Ca^{2+}$ transients in ICC-SM.

## Effects of membrane hyperpolarization on $Ca^{2+}$ transients in ICC-SM

Experiments described above showed that $Ca^{2+}$ transients in ICC-SM depend on voltage-dependent $Ca^{2+}$ influx mechanisms (*Figures 7–10*). The role and contributions of the voltage-dependent channels expressed in ICC-SM ($Ca_V$ 1.3 and $Ca_V$ 1.2 and $Ca_V$ 3.2) was further examined under conditions

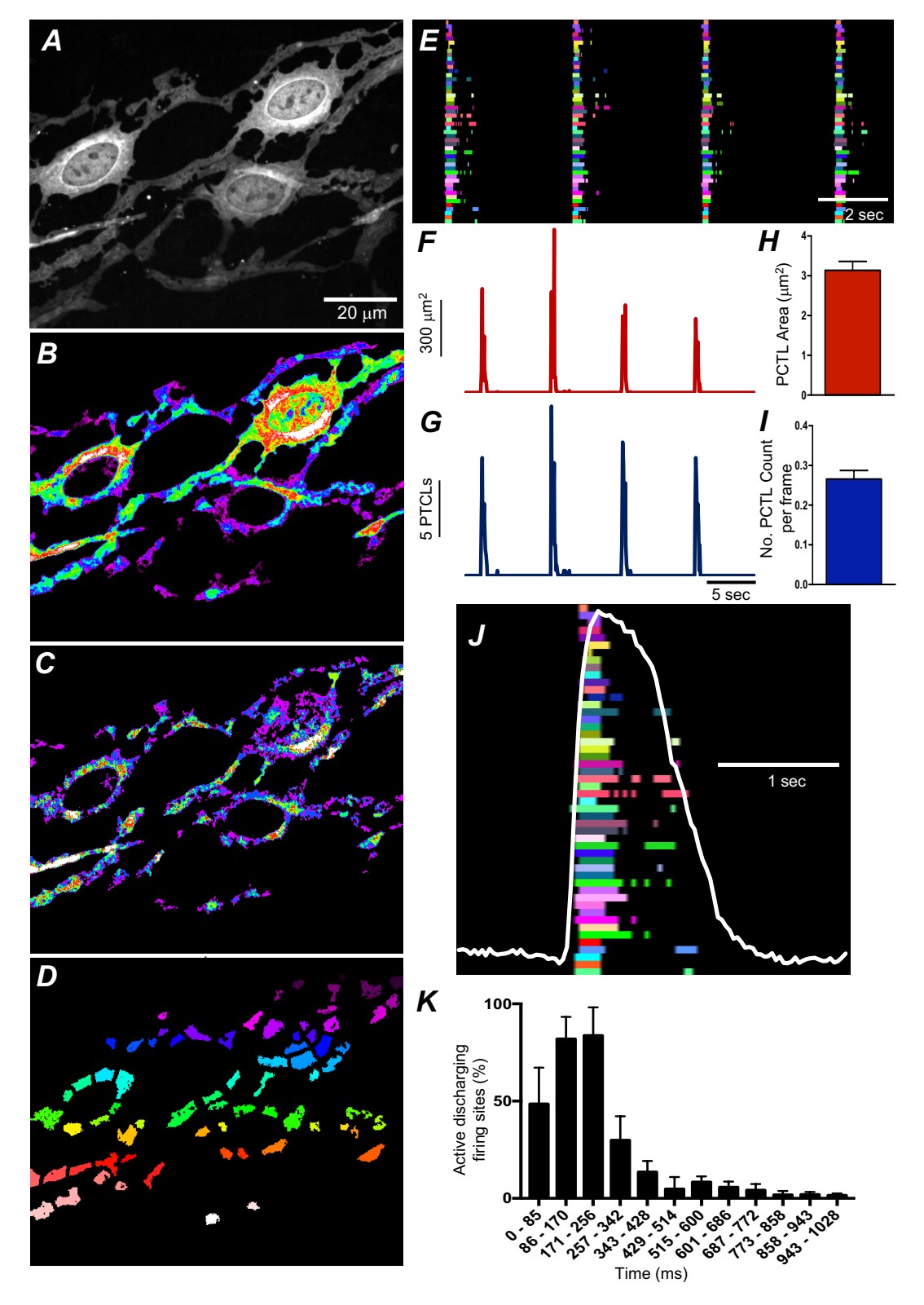

**Figure 6.** Submucosal interstitial cells of Cajal (ICC-SM) Ca$^{2+}$ transient initiation sites. (**A**) Representative image of an ICC-SM network from proximal colon of a Kit-iCre-GCaMP6f mouse at ×60 magnification. (**B**) Heat map of total Ca$^{2+}$ PTCLs generated from the video shown in **A** (see *Video 2*). (**C**) Particles were thresholded temporally to generate a heat map indicating Ca$^{2+}$-firing sites in the ICC-SM. (**D**) Image showing individually color-coded Ca$^{2+}$-firing sites in the field of view (FOV) shown in C. (**E**) The temporal characteristics of each individual, color-coded firing site is displayed as an

*Figure 6 continued on next page*

**Figure 6 continued**

occurrence map, with each 'lane' representing the occurrence of firing PTCLs within each firing site. Activity traces for PTCLs for the duration of recording from the entire FOV are shown in **F and G**. Traces for PTCL area (**F**; red) and PTCL count (**G**; blue) are shown. **H and I** Summary graphs show average PTCL areas and counts for $Ca^{2+}$-firing sites in ICC-SM ($n = 25$). (**J**) One $Ca^{2+}$ wave in ICC-SM (white trace) is expanded and the numerous $Ca^{2+}$ initiation sites that fired during this wave are superimposed. (**K**) Distribution plot showing average percentages of firing sites during a $Ca^{2+}$ wave. Values are calculated for 1 s and plotted in 85 ms bins ($n = 25$). Data graphed as mean ± SEM.

of membrane hyperpolarization induced by activation of $K_{ATP}$ channels. $K_{ATP}$ is functional in colonic SMCs but not in ICC (**Huang et al., 2018**). Therefore, these experiments were performed on preparations in which ICC-SM remained attached to the muscularis. Pinacidil produces hyperpolarization of SMCs via activation of $K_{ATP}$ channels (**Koh et al., 1998**) and the hyperpolarization is expected to conduct to ICC, as they are electrically coupled. Pinacidil (10 µM), a selective $K_{ATP}$ channel agonist, hyperpolarizes murine colonic muscles by ~10 mV (**Kito et al., 2005**; **Koh et al., 1998**; **Kito and Suzuki, 2003**). Pinacidil had no significant change to $Ca^{2+}$ transient firing, except to reduce the duration of CTCs (**Figure 11—figure supplement 1A–L**). $Ca^{2+}$ PTCL area was increased to 130.3 ± 24.3% (**Figure 11—figure supplement 1G**; p *value = 0.24; n = 6*) and PTCL count was increased to 121.8 ± 17.8% (**Figure 11—figure supplement 1H**; p *value = 0.25; n = 6*). The number of firing sites was not affected by pinacidil (**Figure 11—figure supplement 1I**; p *value = 0.22; n = 6*). The effects of pinacidil on global $Ca^{2+}$ transients were evaluated. Pinacidil significantly reduced the duration of global $Ca^{2+}$ transients from 1.78 ± 0.2 s to 0.89 ± 0.1 s (**Figure 11—figure supplement 1J&K**; *n = 6*). Reduction in the duration of $Ca^{2+}$ transients was associated with a tendency for an increase in frequency, but this parameter did not reach significance. $Ca^{2+}$ oscillation frequency under control conditions was 15.1 ± 1.1 cpm and in the presence of pinacidil was 16.5 ± 1.2 cpm (**Figure 11—figure supplement 1J&L**; p *value = 0.45; n = 6*).

The effects of nicardipine were tested in the presence of pinacidil. Under these conditions, nicardipine significantly reduced $Ca^{2+}$ transients in ICC-SM (**Figure 11G&H**; *n = 5*). PTCL area was reduced to 38.5 ± 7.6% (**Figure 11G**; *n = 5*) and PTCL count was reduced to 42.3 ± 9.1% (**Figure 11H**; *n = 5*). In some regards, these results were surprising as membrane potential hyperpolarization might reduce contributions from L-type $Ca^{2+}$ channels ($Ca_V$ 1.2). One explanation is that $Ca_V$ 1.3, which are abundant in ICC-SM and activate at more negative potentials than $Ca_V$ 1.2 (**Xu and Lipscombe, 2001**) may contribute to $Ca^{2+}$ entry at more hyperpolarized potentials. Further

addition of NNC 55–0396 (10 µM) inhibited $Ca^{2+}$ transients in ICC-SM to a greater extent. PTCL area was reduced to 9.0 ± 2% (**Figure 11G**; *n = 5*), and PTCL count was reduced to 11.2 ± 1.9% (**Figure 11H**; *n = 5*). The utilization of two $Ca^{2+}$ conductances with different ranges of voltage-dependent activation for the initiation of CTCs provides a safety factor that insures persistence of pacemaker activity over a broad range of membrane potentials.

Inhibiting voltage-dependent $Ca^{2+}$ channels in the presence of pinacidil unmasked underlying $Ca^{2+}$ transients that occurred more randomly than the clustered transients occurring normally (**Figure 11F&I**). We tabulated the number of $Ca^{2+}$ events in the intervals between CTCs (calculated from a period of 2 s before the onset of a CTC). Underlying $Ca^{2+}$ events were more frequent in the presence of pinacidil and nicardipine and increased again upon addition of NNC 55–0396 (**Figure 11J**; *n = 5*).

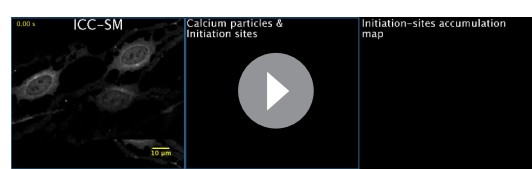

**Video 2.** High spatial resolution ICC-SM $Ca^{2+}$ signals composed of multiple $Ca^{2+}$-firing sites. A video showing subcellular $Ca^{2+}$ transients in ICC-SM at high resolution imaged with a ×60 objective. $Ca^{2+}$ signals were monitored using the genetically encoded $Ca^{2+}$ indicator GCaMP6f. The left panel shows typical stellate-shaped ICC-SM with multiple interconnected processes. The scale bar (yellow) is 10 µm. The middle panel shows the $Ca^{2+}$ particle (PTCL) activity, color coded in blue for raw PTCLs, and the centroids of particles are indicated in purple and green indicates $Ca^{2+}$-firing sites. Note the multiple-site firing of $Ca^{2+}$ transients in ICC-SM. The right panel shows initiation/firing sites accumulation map. The pattern of firing sites $Ca^{2+}$ activity was temporally clustered as $Ca^{2+}$ wave oscillations swept through ICC-SM networks.
https://elifesciences.org/articles/64099#video2

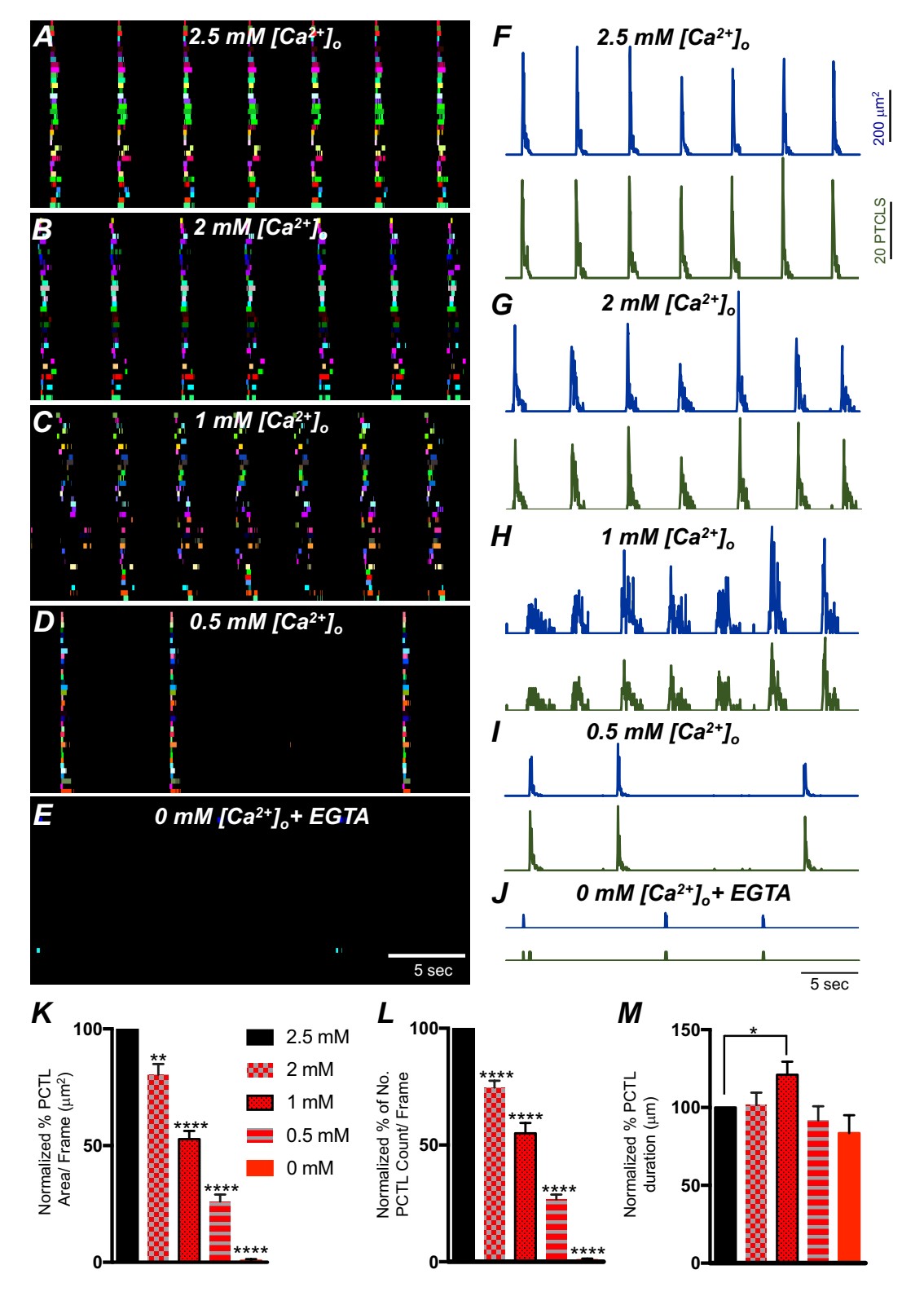

**Figure 7.** The effect of lowering $[Ca^{2+}]_o$ on $Ca^{2+}$ transients in submucosal interstitial cells of Cajal (ICC-SM). (**A**) ICC-SM $Ca^{2+}$ transients and $Ca^{2+}$-firing sites were color coded and plotted as an occurrence map under control conditions with $[Ca^{2+}]_o$ = 2.5 mM. (**B–E**) showing the effects of reducing $[Ca^{2+}]_o$ to 2 mM **B**; 1 mM **C**; 0.5 mM **D** and after $Ca^{2+}$ removal of $[Ca^{2+}]_o$ (0 mM added and final solution buffered with 0.5 mM EGTA) **E**. **F-J** Traces of $Ca^{2+}$ PTCL activity in ICC-SM (PTCL area, blue and PTCL count, green) under control conditions **F** and after reducing $[Ca^{2+}]_o$ to 2 mM **G**; 1 mM **H**; 0.5 mM **I**

*Figure 7 continued on next page*

*Figure 7 continued*

and after removal of $[Ca^{2+}]_o$ as shown in **J**. Summary graphs of $Ca^{2+}$ PTCLs in ICC-SM under control conditions and with reduced $[Ca^{2+}]_o$ in **K** (PTCL area); **L** (PTCL count) and **M** (PTCL duration). Data were normalized to controls and expressed as percentages (%). Significance was determined using one-way ANOVA, * = p<0.1, ** = p<0.01, **** = p<0.0001, *n* = 6. All data graphed as mean ± SEM.

## Contributions of intracellular Ca²⁺ stores and release channels in ICC-SM Ca²⁺ activity

Previous studies have demonstrated that $Ca^{2+}$ signaling in ICC-MY in the small intestine depends not only on $Ca^{2+}$ influx but also on $Ca^{2+}$ release from intracellular stores (*Drumm et al., 2017*). $Ca^{2+}$ release from stores is also critical for generation of pacemaker currents and slow waves (*Ward et al., 2000b*; *Zhu et al., 2015*; *Bayguinov et al., 2007*). The role of $Ca^{2+}$ release mechanisms in $Ca^{2+}$ signaling in ICC-SM was also evaluated. Thapsigargin (1 µM; A SERCA pump antagonist) reduced, but did not block, $Ca^{2+}$ transient firing in ICC-SM (*Figure 12A–D*). PTCL area was reduced to 29 ± 12% (*Figure 12E*; *n* = 6) and PTCL count was reduced to 22 ± 10% (*Figure 12F*; *n* = 6). The number of firing sites was reduced by thapsigargin to 21 ± 11% (*Figure 12G*; *n* = 6). The duration of PTCLs was increased by thapsigargin to 126.7 ± 9% (*Figure 12H*; *n* = 6). Cyclopiazonic acid (CPA, 10 µM), another SERCA antagonist, also reduced $Ca^{2+}$ transient firing (*Figure 12I–L*). PTCL area was reduced to 36.1 ± 8.3% (*Figure 12M*; *n* = 5) and PTCL count was reduced to 29.1 ± 6.3% (*Figure 12N*;

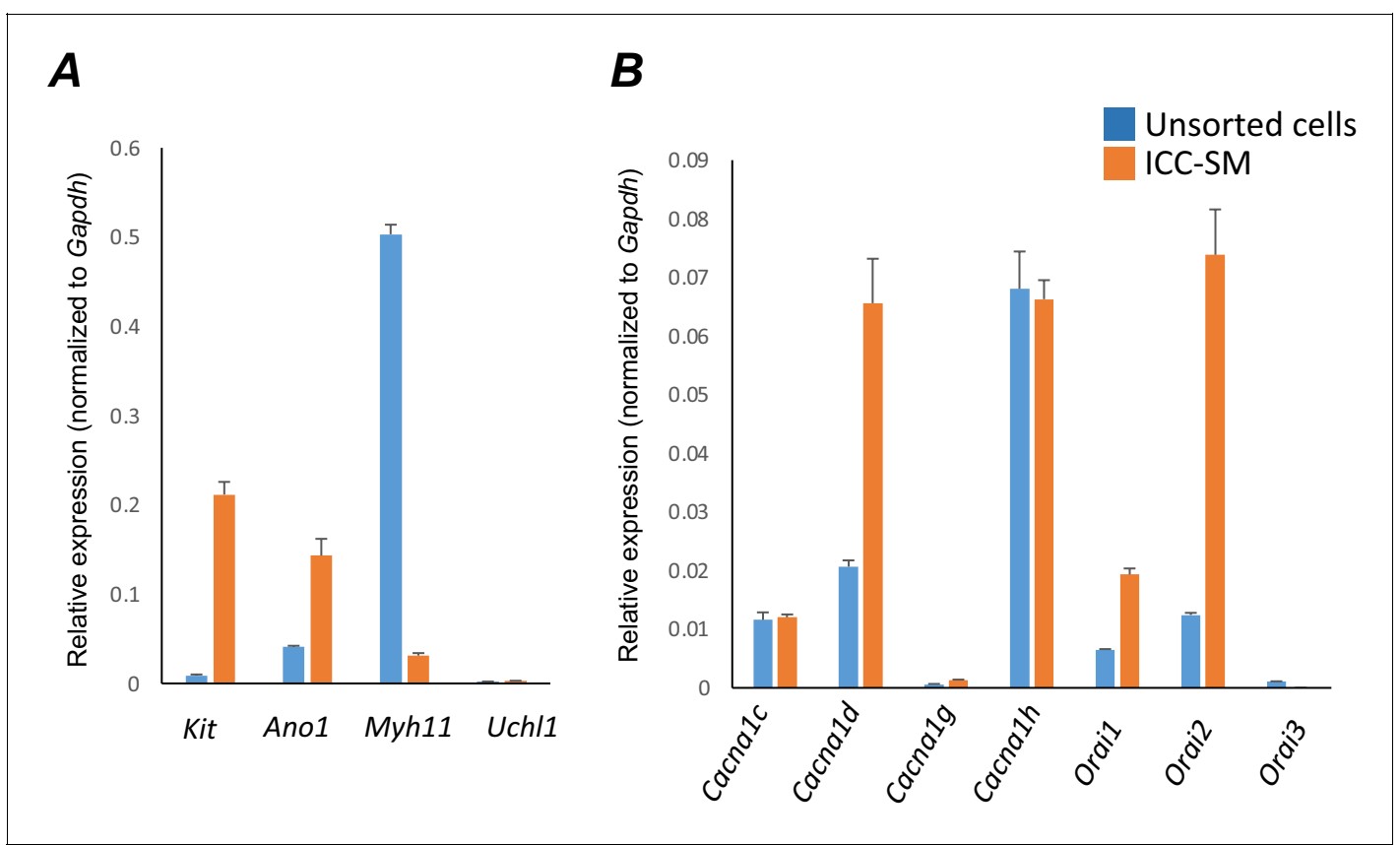

**Figure 8.** Molecular expression of genes related to $Ca^{2+}$ entry channels. (**A**) Relative expression of cellular-specific biomarker genes in submucosal interstitial cells of Cajal (ICC-SM; sorted to purity by FACS) and compared with unsorted cells dispersed from submucosal tissues obtained from Kit$^{+/copGFP}$ mice. Relative expression was determined by qPCR and normalized to *Gapdh* expression. Genes examined were *Kit* (tyrosine kinase receptor, found in ICC), *Ano1* ($Ca^{2+}$-activated Cl⁻ channel), *Uch11* (neural marker encoding PGP 9.5), *Myh11* (smooth muscle myosin). (**B**) Relative expression of major $Ca^{2+}$ entry channels considered most likely to be expressed in colonic ICC from RNAseq of total ICC in murine colon (*Lee et al., 2017*). L-Type $Ca^{2+}$ channels (*Cacna1c* and *Cacna1d*) T-type $Ca^{2+}$ channels (*Cacna1g* and *Cacna1h*) and Store-operated $Ca^{2+}$ entry (SOCE) channels (*Orai1*, *Orai2* and *Orai3*) were evaluated. All data graphed as mean ± SEM (*n* = 4).

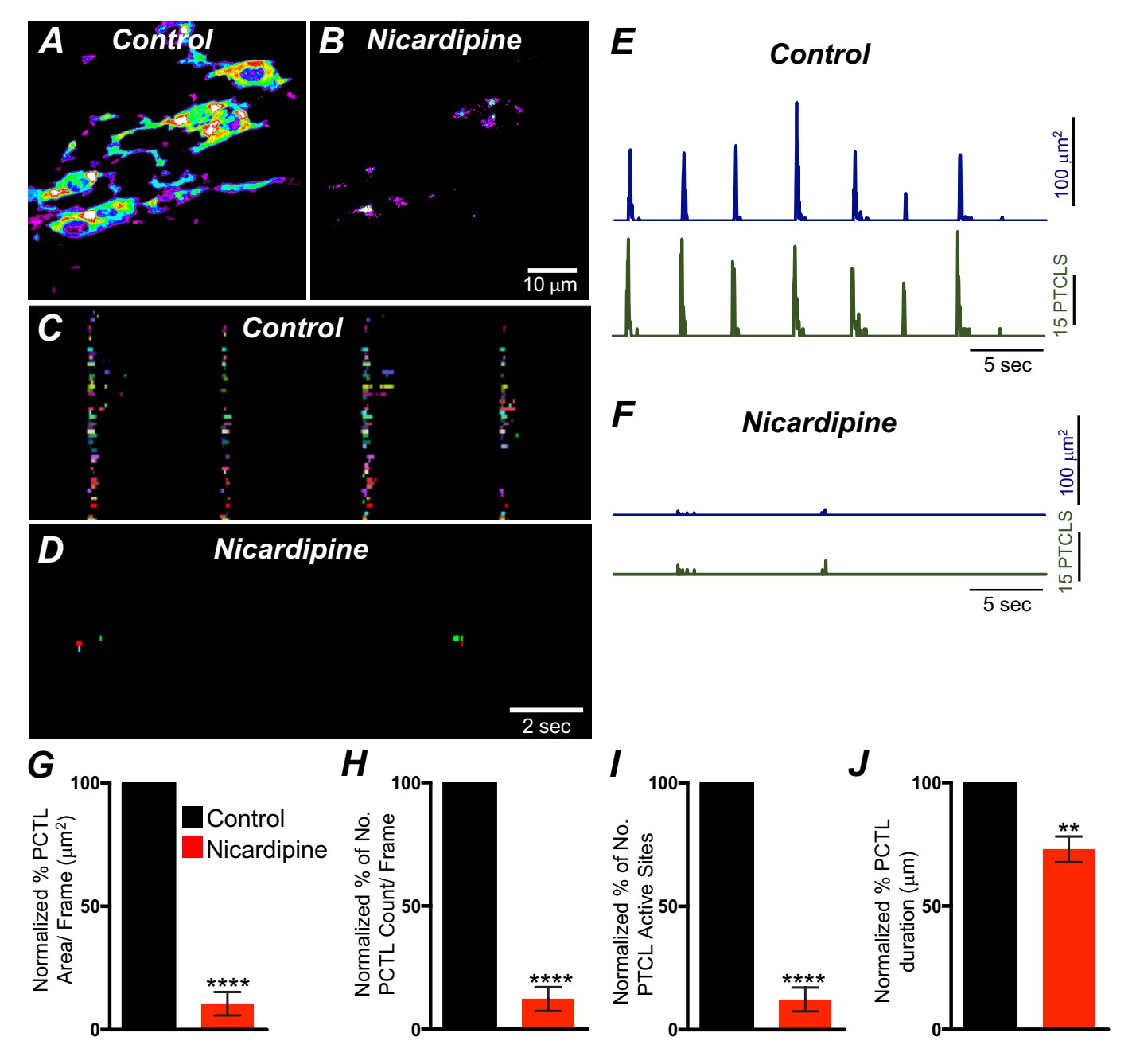

**Figure 9.** L-type Ca$^{2+}$ channel antagonist, nicardipine effects on submucosal interstitial cells of Cajal (ICC-SM) Ca$^{2+}$ transients. **A and B** Representative heat-map images of an ICC-SM network from the proximal colon of a Kit-iCre-GCaMP6f mouse showing active Ca$^{2+}$ PTCLs under control conditions and in the presence of nicardipine (1 μM). Ca$^{2+}$ activity is color coded with warm areas (white, red) representing bright areas of Ca$^{2+}$ fluorescence and cold colors (purple, black) representing dim areas of Ca$^{2+}$ fluorescence. Scale bar is 10 μm in both **A and B**. **C and D** Firing sites showing Ca$^{2+}$ activity in ICC-SM. Firing sites were color coded and plotted as an occurrence map under control conditions and in the presence of nicardipine (1 μM). Traces of firing sites showing PTCL area (**E**; blue) and PTCL count (**E**; green) under control conditions and in the presence of nicardipine; PTCL area (**F**; blue) and PTCL count (**F**; green) show the inhibitory effects of nicardipine on Ca$^{2+}$ transients in ICC-SM. Summary graphs of Ca$^{2+}$ PTCL activity in ICC-SM before and in the presence of nicardipine are shown in **G** (PTCL area/frame), (**H**) (PTCL count/frame), (**I**) the number of PTCL active sites and **J** (PTCL duration). Data were normalized to controls and expressed as percentages (%). Significance determined using unpaired t-test, ** = p<0.01, **** = p<0.0001, *n* = 8. All data graphed as mean ± SEM.

The online version of this article includes the following figure supplement(s) for figure 9:

**Figure supplement 1.** Isradipine effects on Ca$^{2+}$ transients in submucosal interstitial cells of Cajal (ICC-SM).

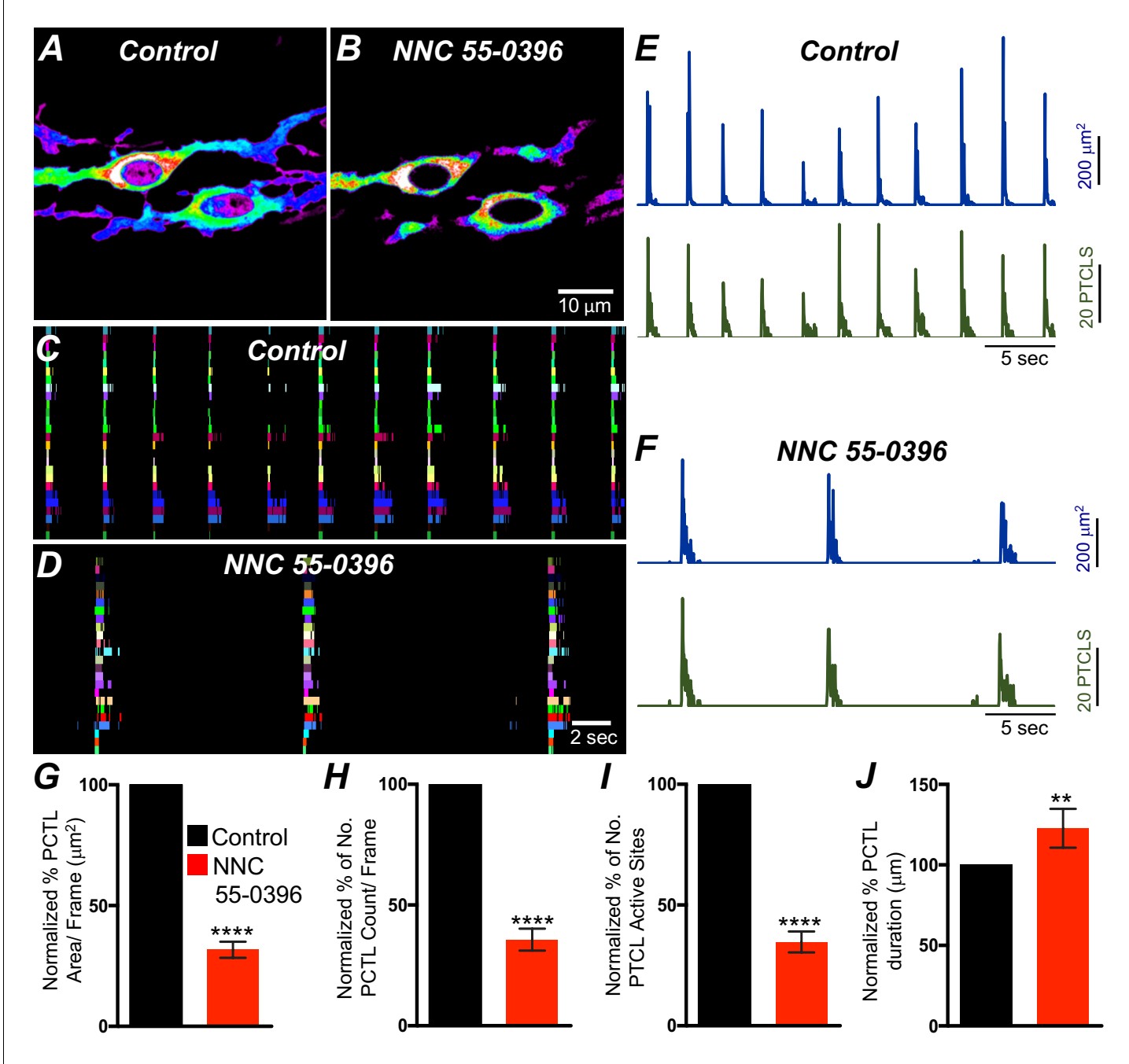

**Figure 10.** T-type Ca$^{2+}$ channel antagonist, NNC 55–0396 effects on ICC-SM Ca$^{2+}$ transients. **A and B** Representative heat-map images of Ca$^{2+}$ transient particles in ICC-SM under control conditions **A** and in the presence of NNC 55–0396 (10 μM) (**B**). Active firing sites were color coded and plotted as an occurrence maps in the ICC-SM network under control **C** and NNC 55–0396 **D** conditions. Plots of Ca$^{2+}$ transient particle activity of ICC-SM in control conditions and in the presence of NNC 55–0396 showing PTCL area (blue) and PTCL count (green) under control conditions **E** and in the presence of NNC 55–0396 **F**. Summary graphs of average percentage changes in PTCL area **G**, PTCL count **H**, the number of PTCL active sites **I** and PTCL duration **J**. Data were normalized to controls and expressed as percentages (%). Significance determined using unpaired t-test, **** = p<0.0001, n = 8. All data graphed as mean ± SEM.

The online version of this article includes the following figure supplement(s) for figure 10:

**Figure supplement 1.** Effects of T-type Ca$^{2+}$ channel antagonists, TTA-A2 and Z-944 on submucosal interstitial cells of Cajal (ICC-SM) Ca$^{2+}$ transients.

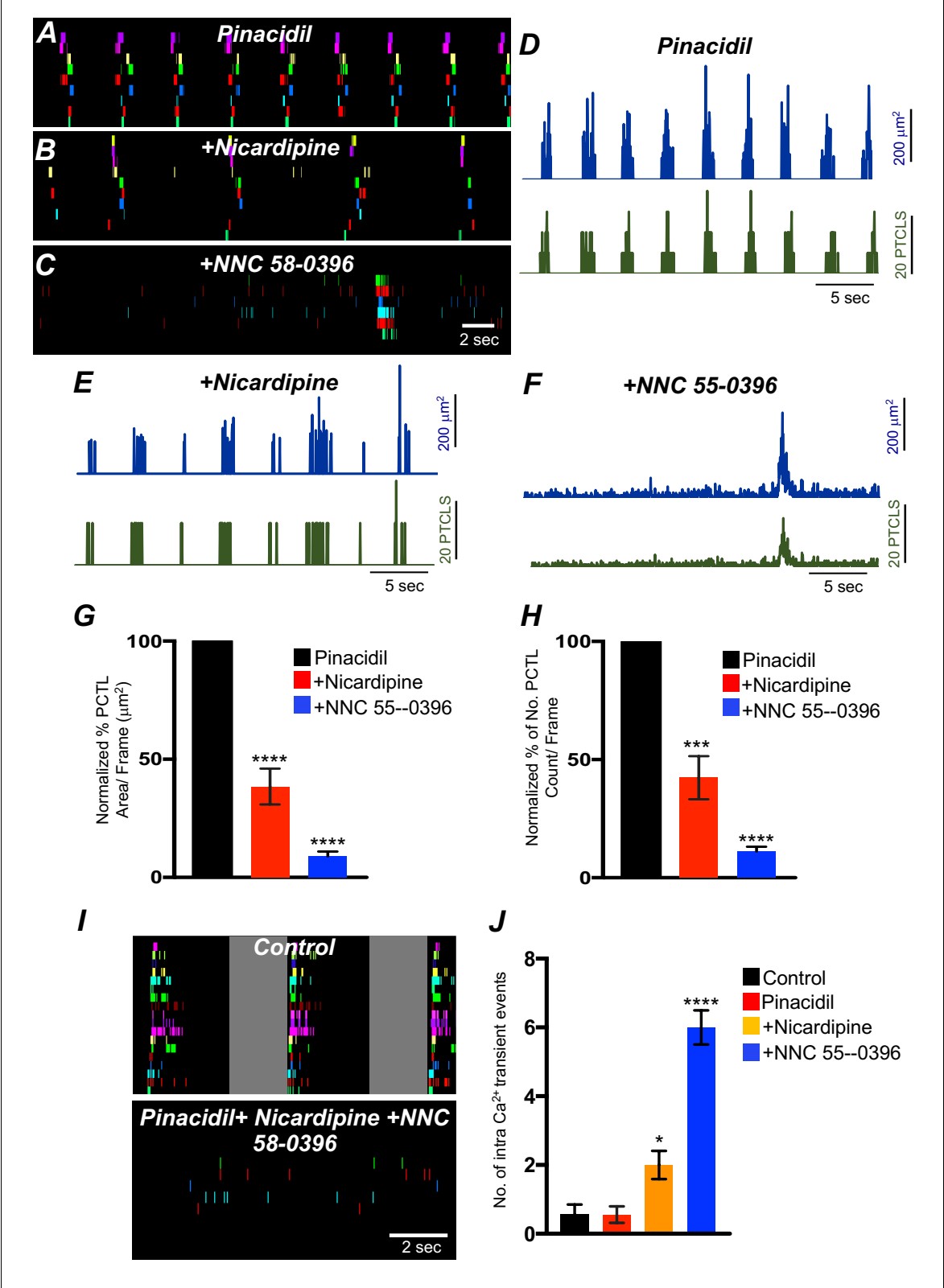

**Figure 11.** The effects of membrane hyperpolarization and voltage-dependent Ca$^{2+}$ entry block on submucosal interstitial cells of Cajal (ICC-SM) Ca$^{2+}$ transients. (**A**) Ca$^{2+}$-firing sites in ICC-SM are color coded and plotted in an occurrence map in the presence of pinacidil (10 μM). (**B**) Shows the effects of nicardipine (1 μM) in the continued presence of pinacidil. (**C**) shows effects of combining nicardipine and NNC 55–0396 (10 μM) in the continued presence of pinacidil. Traces of firing sites PTCL area (blue) and PTCL count (green) under each condition are shown in **D** pinacidil **E** pinacidil and

*Figure 11 continued on next page*

*Figure 11 continued*

nicardipine and **F** combination of pinacidil, nicardipine, and NNC 55–0396. Summary graphs of $Ca^{2+}$ PTCL activity in ICC-SM in the presence of voltage-dependent $Ca^{2+}$ channel antagonists (nicardipine and NNC 55–0396) are shown in **G** (PTCL area) and **H** (PTCL count). (**I**) The number of $Ca^{2+}$ firing events were tabulated during 2 s intervals before the initiation of $Ca^{2+}$ transient clusters in ICC-SM (period of tabulation indicated by the gray box) and summarized in **J** under control conditions, in pinacidil, in pinacidil and nicardipine and in a combination of pinacidil, nicardipine, and NNC 55–0396. Data were normalized to controls and expressed as percentages (%) in **G** and **H**. Significance determined using one-way-ANOVA, * = p<0.1, ** = p<0.01, *** = p<0.001 **** = p<0.0001, *n* = 5. All data graphed as mean ± SEM.

The online version of this article includes the following figure supplement(s) for figure 11:

**Figure supplement 1.** The effects of pinacidil of $Ca^{2+}$ transients in submucosal interstitial cells of Cajal (ICC-SM).

---

*n* = 5). The number of firing sites was reduced by CPA to 56 ± 9.5% (*Figure 12O*; *n* = 5). There was no significant change in the duration of PTCLs in the presence of CPA (*Figure 12P*; *n* = 5).

ER release channels RyRs and $IP_3Rs$ amplify and sustain $Ca^{2+}$ signaling via direct localized release of $Ca^{2+}$ or $Ca^{2+}$-induced $Ca^{2+}$ release (CICR) (*Kaßmann et al., 2019*; *Vierra et al., 2019*; *Amberg et al., 2006*; *Takeda et al., 2011*; *Fill and Copello, 2002*; *Dupont and Goldbeter, 1994*; *Thomas et al., 1996*; *Berridge, 2009*). Contributions from RyRs and $IP_3Rs$ to $Ca^{2+}$ transients in ICC-SM were therefore investigated. Ryanodine (100 µM) significantly reduced $Ca^{2+}$ event firing in ICC-SM (*Figure 13A–F*). PTCL area was reduced to 76.1 ± 1.6% (*Figure 13G*; *n* = 4) and PTCL count was reduced to 80.9 ± 2.7% (*Figure 13H*; *n* = 4). The number of firing sites was also reduced by ryanodine to 75.7 ± 2.1% (*Figure 13I*; *n* = 4). There was no significant change in the duration of PTCLs in the presence of ryanodine (*Figure 13J*; *n* = 5). We also noted that the greatest effects of ryanodine on $Ca^{2+}$ transients occurred after the first ~300 ms of CTCs (*Figure 13K&L*; *n* = 4). Thus, ryanodine shortens the durations of the CTCs. $Ca^{2+}$ release mechanisms via RyRs contribute to the overall patterns of $Ca^{2+}$ waves in ICC-SM, as shown by the distribution plots of average percentages of firing sites during a $Ca^{2+}$ wave (*Figure 13L*; *n* = 4).

Xestospongin C (10 µM; An $IP_3R$ antagonist) also reduced $Ca^{2+}$ events in ICC-SM (*Figure 13—figure supplement 1A–I*; *n* = 4). PTCL area was significantly reduced to 78 ± 5.8% (*Figure 13—figure supplement 1G*; *n* = 4) and although both PTCL count and number were reduced to 75.3 ± 10.5% and 74.8 ± 9.3%, respectively, these effects did not reach statistical significance (PTCL count: p value = 0.08 and PTCL number: p value = 0.06; *Figure 13—figure supplement 1H&I*; *n* = 4). Xestospongin C displayed inhibitory effects similar to ryanodine; most of the inhibition of $Ca^{2+}$ events occurred after the first ~400 ms of CTCs (*Figure 13—figure supplement 1J&K*; *n* = 4). Thus, $Ca^{2+}$ release via $IP_3Rs$ also contributes to the overall pattern of $Ca^{2+}$ waves in ICC-SM, as shown by the distribution plots of average percentages of firing sites during a $Ca^{2+}$ wave (*Figure 13—figure supplement 1K*; *n* = 4).

2-APB (100 µM) and tetracaine were also tested (100 µM; *Figure 13—figure supplement 2A&B*; *n* = 5) as secondary tests of the contributions of $IP_3Rs$ and RyRs in CTCs. 2-APB reduced $Ca^{2+}$ PTCL area to 61.1 ± 14.5% (*Figure 13—figure supplement 2 Ai*; *n* = 5) and reduced PTCL count to 58 ± 13.2% (*Figure 13—figure supplement 2 Aii*; *n* = 5). The number of firing sites was also reduced by 2-APB to 63.7 ± 11.3% (*Figure 13—figure supplement 2 Aiii*; *n* = 5). Tetracaine reduced $Ca^{2+}$ PTCL area to 85 ± 5.1% (*Figure 13—figure supplement 2 Bi*; *n* = 5) and reduced PTCL count to 69.5 ± 7.8% (*Figure 13—figure supplement 2 Bii*; *n* = 5). The number of firing sites was also reduced by tetracaine to 71.3 ± 5.9% (*Figure 13—figure supplement 2 Biii*; *n* = 5).

The effects of 2-APB could be non-specific and may include effects on store-operated $Ca^{2+}$ entry channels (SOCE; e.g. by blocking Orai channels). Previous studies have shown SOCE to be important for maintenance of $Ca^{2+}$ stores and sustaining $Ca^{2+}$ release from the ER (*Zheng et al., 2018*; *Lyfenko and Dirksen, 2008*; *Trebak et al., 2013*; *Chen and Sanderson, 2017*; *Putney, 2011*; *Prakriya and Lewis, 2015*). ICC-SM express Orai channels (*Orai1* and Orai*2*; *Figure 8B*), so the role of SOCE in maintenance of $Ca^{2+}$ transients was examined using an Orai antagonist. GSK 7975A (10 µM; An Orai antagonist) reduced the firing frequency of CTCs (*Figure 14A&B*). Firing site occurrence (*Figure 14C&D*) and PTCL counts and areas were reduced (*Figure 14E&F*). $Ca^{2+}$ PTCL area was reduced to 42.4 ± 9.4% (*Figure 14E–G*; *n* = 7), and PTCL count was reduced to 48 ± 7% (*Figure 14H*; *n* = 7). The number of firing sites was also inhibited by GSK 7975A to 47.5 ± 4.1% (*Figure 14I*; *n* = 7). There was no significant change in the duration of PTCLs in the presence of GSK 7975A (*Figure 14J*; *n* = 7).

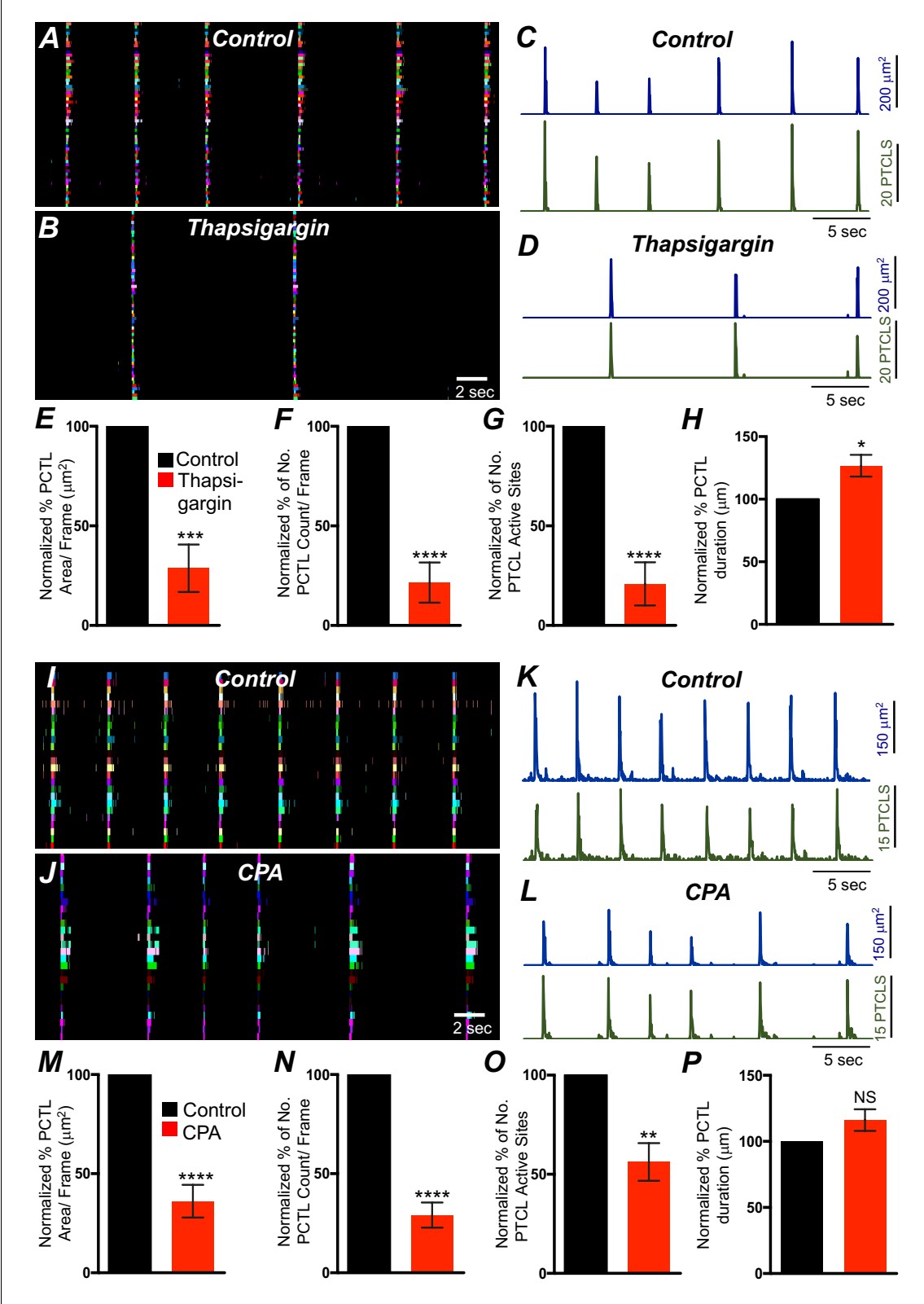

**Figure 12.** Intracellular Ca$^{2+}$ stores contributions in submucosal interstitial cells of Cajal (ICC-SM) Ca$^{2+}$ transients firing. (A) Ca$^{2+}$ activity of firing sites in ICC-SM are color coded and plotted in occurrence maps under control conditions and in the presence of thapsigargin (1 μM) (B). Traces of firing sites PTCL area (C; blue) and PTCL count (C; green) under control conditions and in the presence of thapsigargin PTCL area (D; blue) and PTCL count (D; green). Scale bars in C applies to traces in D. Summary graphs of Ca$^{2+}$ PTCL activity in ICC-SM in the presence of thapsigargin are shown in E

*Figure 12 continued on next page*

*Figure 12 continued*

(PTCL area), (F) (PTCL count), (G) the number of PTCL active sites and H (PTCL duration; $n = 6$). CPA (SERCA pump inhibitor) reduced transients compared to control as shown in occurrence maps of firing sites I and J and $Ca^{2+}$ activity traces K and L. Summary graphs of $Ca^{2+}$ PTCL activity in ICC-SM in the presence of CPA are shown in M (PTCL area), (N) (PTCL count), (O) the number of PTCL active sites and P (PTCL duration; $n = 5$). Data were normalized to controls and expressed as percentages (%). Significance determined using unpaired t-test, * = $p<0.1$, ** = $p<0.01$, *** = $p<0.001$ **** = $p<0.0001$. All data graphed as mean ± SEM.

## Discussion

This study characterized $Ca^{2+}$ transients responsible for the pacemaker function of ICC-SM that contributes to contractile patterns in colonic motility. The sequence of activation from ICC-SM to colonic SMCs was quantified using two optogenetic sensors, expressed specifically in ICC or SMCs. Correlation analysis demonstrated a 1:1 relationship between CTCs in ICC-SM and $Ca^{2+}$ signaling and contractile responses in SMCs. The CTCs in ICC-SM consisted of ~2 s bursts of activity from multiple sites within cells. Organization of the $Ca^{2+}$ transients into clusters was due to voltage-dependent $Ca^{2+}$ entry that appeared to be due to activation of both high-voltage activated $Ca^{2+}$ channels (L-type encoded by *Cacna1c* and *Cacna1d* in ICC-SM) and low-voltage activated $Ca^{2+}$ channels (T-type encoded by *Cacna1h* and possibly *Cacna1g* in ICC-SM). A portion of the $Ca^{2+}$ transients making up CTCs were due to $Ca^{2+}$ entry, as the earliest $Ca^{2+}$ transients resolved in a cluster were not blocked by thapsigargin, CPA or antagonists of ryanodine and $IP_3$ receptors. The earliest events were blocked by nicardipine, suggesting that L-type $Ca^{2+}$ channels are the dominant $Ca^{2+}$ entry pathway at basal resting potentials. $Ca^{2+}$ transients in ICC-SM were not as sensitive to block by STIM and Orai, as in other ICC (*Zheng et al., 2018*; *Drumm et al., 2019a*), However, the Orai antagonist reduced the frequency of CTCs and may have blocked these events completely with longer treatment periods. Our data show that $Ca^{2+}$ entry is fundamental in ICC-SM $Ca^{2+}$ transients. In fact, localized $Ca^{2+}$ influx via clusters of L-type $Ca^{2+}$ channels may cause localized elevation in $[Ca^{2+}]_i$ and activation of ANO1 channels directly. Localized $Ca^{2+}$ entry events through L-type $Ca^{2+}$ channels have been termed $Ca^{2+}$sparklets, and these events could be involved in $Ca^{2+}$ influx (*Santana et al., 2008*; *Sonkusare et al., 2012*; *Navedo and Amberg, 2013a*) and initiation of pacemaker activity in ICC-SM. However, the decrease in the frequency of CTCs by manipulations to reduce stored $Ca^{2+}$ also suggest an important role for $Ca^{2+}$ release in pacemaker activity, perhaps resulting from coupling between $Ca^{2+}$ entry and CICR (*Navedo and Santana, 2013b*).

In this study, we developed a new preparation in which ICC-SM adherent to the submucosa was used to allow very high-resolution imaging without complications from muscular contractions. In situ preparations of this type in which the natural structure and connectivity between ICC are maintained may be valuable for future studies of cellular mechanisms responsible for pacemaker activity and factors that regulate or degrade pacemaker activity in pathophysiological conditions.

The pacemaker function of ICC-SM was demonstrated in a novel manner by simultaneous two-color optogenetic imaging with green (GCaMP6f) and red (RCaMP1.07) $Ca^{2+}$ sensors expressed in ICC and SMCs, respectively. Imaging in this manner revealed the sequence of activation in ICC-SM and SMCs, showing clearly the frequency, onset and duration of $Ca^{2+}$ transients in ICC-SM, the spatial spread of $Ca^{2+}$ transients in ICC-SM networks, the development of $Ca^{2+}$ transients in SMCs and tissue displacement (i.e. an optical indicator of muscle contraction). Correlation analysis demonstrated the coherence of these events. $Ca^{2+}$ transients, lasting for about 2 s, propagated without decrement through networks of ICC-SM and preceded and likely initiated $Ca^{2+}$ signaling and contractions in SMC, as was also suggested by intracellular microelectrode recordings from cells along the innermost surface of canine colonic muscles (*Sanders et al., 1990*). ICC-SM are electrically coupled to each other and the network of ICC-SM is coupled to SMCs via gap junctions, providing a means of electrical communication. $Ca^{2+}$ transients initiate depolarization of ICC-SM via activation of ANO1 channels, depolarizing currents conduct to SMCs, and depolarization of SMCs activates $Ca^{2+}$ entry, by increasing the open probability of L-type $Ca^{2+}$ channels, and excitation-contraction coupling. Depolarizing signals from multiple ICC-SM influence the excitability of SMC, and the overall depolarization driving SMCs results from the summation of activity from the ICC-SM network.

Slow waves with characteristics similar to those found in the stomach and small intestine (i.e. relatively fast upstroke depolarization and a plateau phase) are generated along the submucosal surface

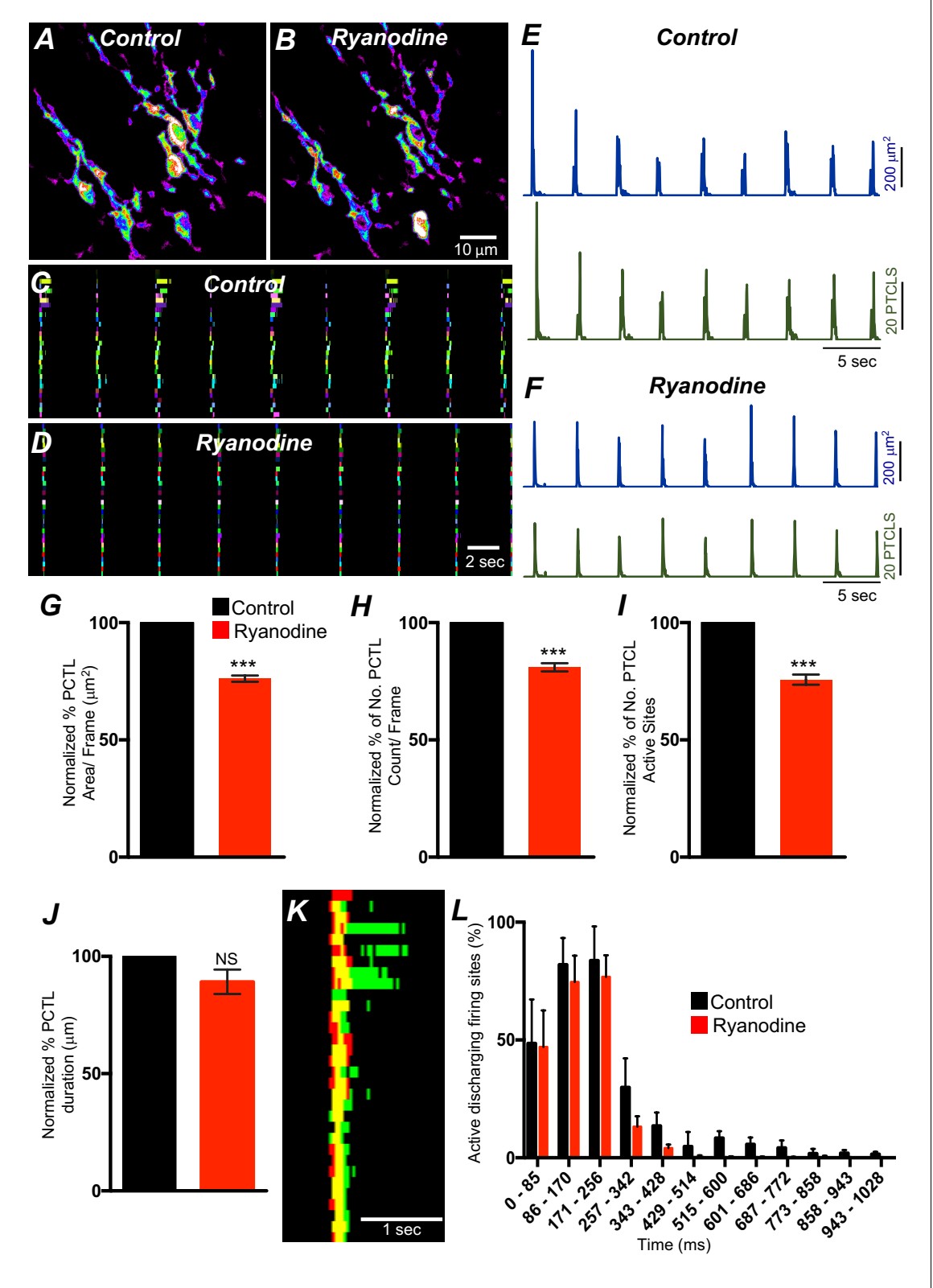

**Figure 13.** Ryanodine receptors (RyRs) contributing to Ca$^{2+}$ release in submucosal interstitial cells of Cajal (ICC-SM). (**A**) Representative heat-map image of an ICC-SM network from proximal colon showing total active Ca$^{2+}$ PTCLs under control conditions and in the presence of ryanodine (100 μM) (**B**). **C** and **D** Ca$^{2+}$-firing sites are color coded and plotted in occurrence maps showing the effect of the ryanodine (100 μM), on Ca$^{2+}$ transient clusters (CTCs) in ICC-SM. Traces of firing sites PTCL area (**E**; blue) and PTCL count (**E**; green) under control conditions and in the presence of ryanodine, PTCL

*Figure 13 continued on next page*

*Figure 13 continued*

area (F; blue) and PTCL count (F; green). Summary graphs of $Ca^{2+}$ PTCL activity in ICC-SM in the presence of ryanodine are shown in **G** (PTCL area), (**H**) (PTCL count), (**I**) the number of PTCL active sites and **J** (PTCL duration; *n* = 4). (**K**) Overlaid occurrence maps showing $Ca^{2+}$ firing during control conditions (all firing sites are in green) and in the presence of ryanodine (all firing sites are in red). Note how ryanodine shortened the duration of the total $Ca^{2+}$ transient cluster (CTC). (**L**) Distribution plot of average percentages of firing sites during a $Ca^{2+}$ wave, calculated for 1 s duration and plotted in 85 ms bins showing that ryanodine mainly blocked $Ca^{2+}$ transients occurring after the first 257 ms intervals (*n* = 4). Data were normalized to controls and expressed as percentages (%). Significance determined using unpaired t-test, *** = p<0.001. All data graphed as mean ± SEM.

The online version of this article includes the following figure supplement(s) for figure 13:

**Figure supplement 1.** IP$_3$ receptors (IP$_3$Rs) contribution to $Ca^{2+}$ transients in submucosal interstitial cells of Cajal (ICC-SM).

**Figure supplement 2.** Effects of 2-APB and Tetracaine on submucosal interstitial cells of Cajal (ICC-SM) $Ca^{2+}$ transients.

of the CM layer in the colon (*Smith et al., 1987a*). It was discovered that peeling the submucosa from the innermost surface of CM blocked slow waves (*Durdle et al., 1983*). While the authors of that study thought this tissue was mostly connective tissue with possibly some adherent SMCs, it became clear that a population of pacemaker cells, ICC-SM, are present along the submucosal surface (*Berezin et al., 1988*; *Ward et al., 1991*). In the present study, we found that ICC-SM and the networks they form are preserved and remain functionally similar in isolated submucosal tissues to ICC-SM attached to the muscularis. It should be noted that ICC-SM were more adherent to CM in the distal colon, and it was more difficult to obtain ICC-SM/submucosal preparations from that region. Submucosal tissues with adherent ICC-SM were used in the current study to eliminate movement artifacts generated by muscle contractions that plague high-resolution $Ca^{2+}$ imaging in most smooth muscle tissues.

While the frequency of pacemaker activity was relatively stable over time in a given preparation, the sequence of activation of individual ICC-SM within the network varied as a function of time, as previously observed in gastric (*Hennig et al., 2004*) and small intestinal (*Park et al., 2006*) ICC-MY networks. What appeared as global $Ca^{2+}$ transients in low resolution imaging partitioned into clusters of localized $Ca^{2+}$ transients when viewed with a ×60 objective at 30 frames/s or at higher acquisition speeds. Summation of the clustered events reproduced the frequency and duration of the $Ca^{2+}$ waves observed at low resolution. Multiple firing sites, averaging ~8 per cell, were identified. This pattern of clustered $Ca^{2+}$ transients (CTCs) was also observed in ICC-MY of the small intestine, the pacemaker cells in that region (*Drumm et al., 2017*). Organization of $Ca^{2+}$ transients into clusters in ICC-SM was dependent upon voltage-dependent $Ca^{2+}$ entry, and our data revealed that in contrast to ICC-MY of the small intestine, $Ca^{2+}$ entry by both dihydropyridine-sensitive and insensitive mechanisms contributes to clustering and propagation of $Ca^{2+}$ waves in intact networks. Nicardipine and isradipine reduced the occurrence of CTCs dramatically, and the T-channel antagonists, NNC 55–0396, TTA-A2, and Z-944 also reduced the occurrence and disordered the $Ca^{2+}$ transients. *Cacna1c*, *Cacna1d*, and *Cacna1h* were expressed in purified ICC-SM, and the presence and function of these channels can explain the pharmacological observations. Channels resulting from *Cacna1d* (encoding Ca$_V$α1D) activate at relatively hyperpolarized membrane potentials and their currents are partially inhibited by dihydropyridines (~50–70% of current density block) in comparison to *Cacna1c* gene products (Ca$_V$α1C) (*Xu and Lipscombe, 2001*; *Bell et al., 2001*), but isradipine blocks Ca$_V$α1C and Ca$_V$α1D equally (*Anekonda et al., 2011*; *Berjukow et al., 2000*; *Scholze et al., 2001*). The fact that isradipine had no greater effect on the occurrence of clustered $Ca^{2+}$ transients than nicardipine suggests that the L-type component of $Ca^{2+}$ entry may be carried primarily by Ca$_V$α1C channels. We have observed relatively robust expression of *Cacna1d* in a variety of ICC in mice (*Drumm et al., 2017*; *Lee et al., 2017*); however, the function of these channels has not been identified specifically.

Having three independent voltage-dependent $Ca^{2+}$ conductances with different voltage-dependent properties coordinate clustering of $Ca^{2+}$ transients provides a safety factor for preservation of pacemaker activity over a broad range of membrane potentials. In spite of overarching changes in membrane potential that might influence the availability of ion channels with narrow voltage-ranges, the broader range of activation potentials offered by expression and function of both L-type and T-type $Ca^{2+}$ channels might protect against voltage-dependent inhibition of pacemaker activity. L-type channels are activated at less polarized potentials than T-type channels (*Nowycky et al., 1985*). Thus, a factor producing tonic hyperpolarization of the SIP syncytium (e.g. purinergic inhibitory neurotransmission; *Gallego et al., 2008*; *Hwang et al., 2012*) may tend to switch the dominant

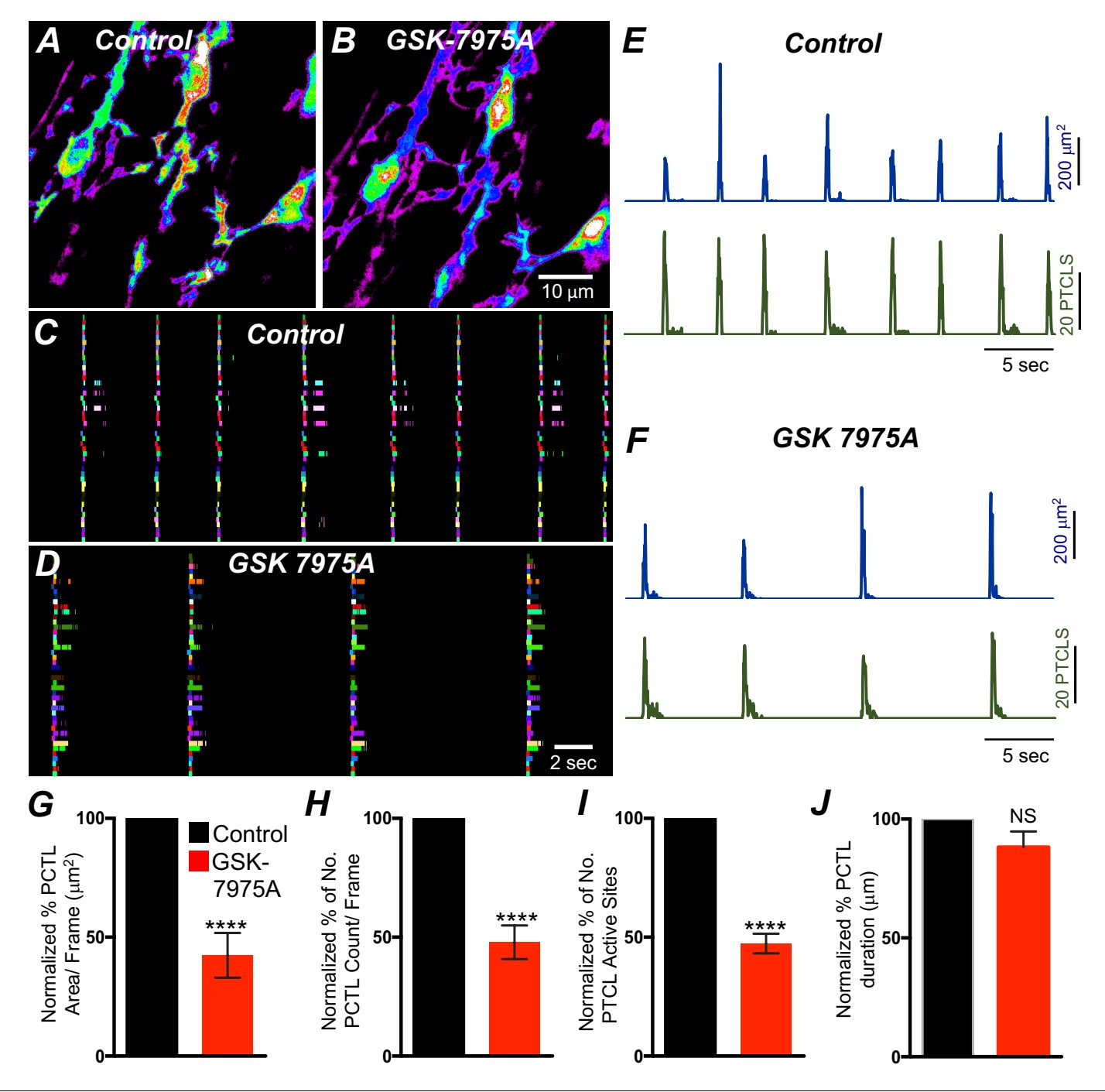

**Figure 14.** SOCE role in maintaining submucosal interstitial cells of Cajal (ICC-SM) Ca$^{2+}$ transients. (**A**) Representative Heat-map image of an ICC-SM network showing total active Ca$^{2+}$ PTCLs under control conditions and in the presence of GSK-7975A (10 μM, for 20 min) B. **C and D** Ca$^{2+}$-firing sites are color coded and plotted in occurrence maps showing the effect of the SOCE channel antagonist, GSK-7975A (100 μM), on ICC-SM Ca$^{2+}$ transients. Traces of PTCL area (**E**; blue) and PTCL count (**E**; green) under control conditions and in the presence of GSK-7975A, PTCL area (**F**; blue) and PTCL count (**F**; green). Summary graphs of Ca$^{2+}$ PTCL activity in ICC-SM in the presence of GSK-7975A are shown in **G** (PTCL area), (**H**) (PTCL count), (**I**) the number of PTCL active sites, and **J** (PTCL duration; *n* = 7). Significance determined using unpaired t-test, **** = p<0.0001. All data graphed as mean ± SEM.

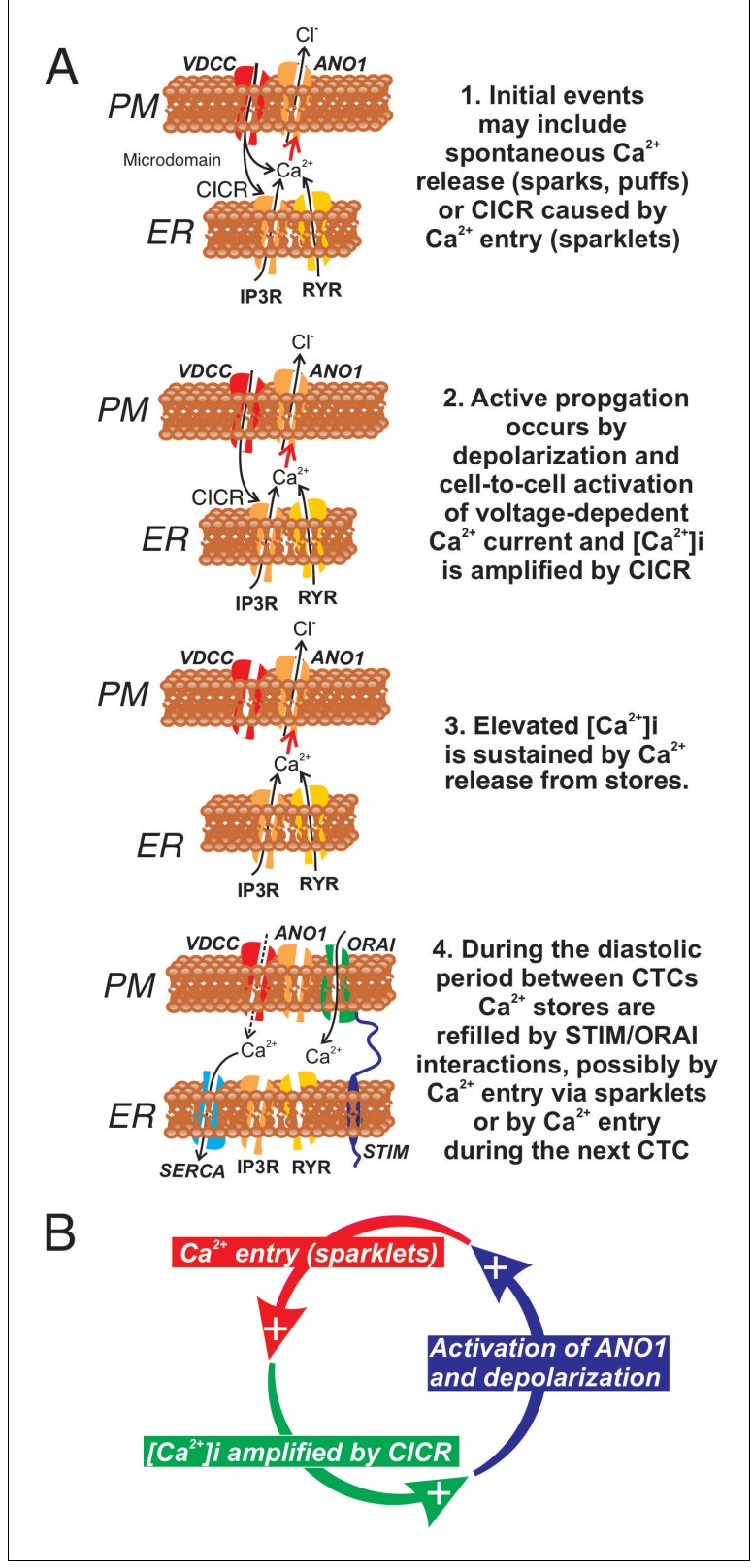

**Figure 15.** Role of voltage-dependent Ca$^{2+}$ entry in the pacemaker function of submucosal interstitial cells of Cajal (ICC-SM). (**A**) Shows segments of plasma membrane (PM) and endoplasmic reticulum membrane (ER) that form PM-ER junctions and microdomains. At least three types of voltage-dependent Ca$^{2+}$ channels (VDCC) are expressed in ICC-SM, Ca$_V$1.2, Ca$_V$1.3, and Ca$_V$3.2. These conductances, with voltage-dependent activation and

*Figure 15 continued on next page*

*Figure 15 continued*

inactivation properties spanning a broad range of negative potentials, insure maintenance of pacemaker activity under conditions of hyperpolarization or depolarization in ICC-SM. Pacemaker activity (1. Initial events) in ICC-SM could be due to spontaneous release of $Ca^{2+}$ from stores in the ER and utilize either $IP_3R$ or RYR receptors or both ($Ca^{2+}$ sparks and puffs). However, our data cannot rule out the possibility that transient openings of voltage-dependent $Ca^{2+}$ channels (sparklets) and amplification of $Ca^{2+}$ in microdomains by CICR constitute the initial events of pacemaker activity. In this case, $Ca^{2+}$ release from stores is not the primary pacemaker event but a secondary response to $Ca^{2+}$ entry. Inhibition of $Ca^{2+}$ release from stores would lead to reduced probability of CICR and decrease the frequency of CTCs. Our hypothesis is that $Ca^{2+}$ entry and/or release from stores activates $Ca^{2+}$-dependent $Cl^-$ current due to ANO1 channels in the plasma membrane. Active propagation between cells in interstitial cells of Cajal (ICC) networks (Phase 2) was inhibited by blocking voltage-dependent $Ca^{2+}$ channels. Active propagation may also require or depend upon amplification of $Ca^{2+}$ in microdomains by CICR. The duration of $Ca^{2+}$ entry is likely to be brief due to voltage-dependent inactivation of L- and T-type $Ca^{2+}$ channels. The duration of CTCs appears to be enhanced by CICR (Phase 3). Our data show that the duration of CTCs is reduced by several manipulations known to inhibit $Ca^{2+}$ release from stores. In Phase four store reloading may occur by multiple mechanisms and may include: (i) transient $Ca^{2+}$ entry via sparklets, (ii) activation of SOCE via STIM/ORAI interactions, and (iii) the increase in $Ca^{2+}$ entry that occurs via depolarization and activation of $Ca^{2+}$ entry at the onset of each CTC. (B) A novel hypothesis emerges from this study suggesting that the pacemaker mechanism in non-voltage-clamped cells includes a cyclical, positive-feedback phenomenon that may be responsible for initiation of CTCs and relies on: (i) $Ca^{2+}$ entry through voltage-dependent $Ca^{2+}$ channels. Openings of clusters of these channels would generate sparklets; (ii) $Ca^{2+}$ entry initiates CICR which amplifies $[Ca^{2+}]_i$ within microdomains; (iii) the rise in $[Ca^{2+}]_i$ activates ANO1 channels in the PM causing depolarization; (iv) depolarization enhances the open probability of voltage-dependent $Ca^{2+}$ channels, increasing $Ca^{2+}$ entry. This cycle creates positive feedback for $Ca^{2+}$ entry, clustering of localized $Ca^{2+}$ transients due to $Ca^{2+}$ entry during the first 350–450 ms of CTCs and development of slow wave depolarizations in ICC-SM.

The online version of this article includes the following figure supplement(s) for figure 15:

**Figure supplement 1.** Schematic showing how $Ca^{2+}$ signaling in the pacemaker function of submucosal interstitial cells of Cajal (ICC) is coupled to smooth muscle cells (SMCs).

---

voltage-dependent $Ca^{2+}$ entry mechanism from L-type to T-type $Ca^{2+}$ channels. This concept was demonstrated by the decreased inhibitory effects of nicardipine and increased effects of NNC 55–0396 on $Ca^{2+}$ transients after exposure of tissues to pinacidil. The opposite might be true if the SIP syncytium experiences a depolarizing influence (e.g. neurogenic or humerogenic).

Pinacidil hyperpolarizes colonic muscles through activation of $K_{ATP}$ channels in SMCs (*Koh et al., 1998*). This compound increased the frequency and decreased the duration of CTCs. These results are consistent with the effects of pinacidil on electrical pacemaker activity in the small intestine where it increases the $dV/dt_{max}$ of the upstroke depolarization and decreases the duration of slow waves (*Kito et al., 2005*). The increase in frequency may have been due to reduced inactivation and increased availability of $Ca_V\alpha1D$ and T-type channels ($Ca_V\alpha1H$) at more hyperpolarized potentials. The decrease in the duration of CTCs may be due to a relative shift in the importance of T-type vs. L-type $Ca^{2+}$ channels with hyperpolarization. In the presence of pinacidil, NNC 55–0396 had increased antagonistic effects on CTCs. $Ca^{2+}$ currents via T-type channels inactivate rapidly, whereas L-type channel inactivation is slower and incomplete (*Hirano et al., 1989*; *Tseng and Boyden, 1989*; *Xu and Best, 1992*). Thus, the $Ca^{2+}$ entry period for T-type currents is likely to be more transient than with L-type currents. Channel density in proximity to $Ca^{2+}$ release channels may also affect the degree of coupling between $Ca^{2+}$ entry and CICR, but, as yet, little is known about the structure and functional components of microdomains in ICC.

The importance of $Ca^{2+}$ entry as the primary means of activation and organization of pacemaker activity in ICC-SM was shown by the discoordination of $Ca^{2+}$ transients when extracellular $Ca^{2+}$ was decreased and the incomplete effects of thapsigargin and CPA on $Ca^{2+}$ transients. We noted tight clustering of $Ca^{2+}$ transients at 2.5 and 2.0 mM $[Ca^{2+}]_o$ but the tightness of the CTCs disassociated when the driving force for $Ca^{2+}$ entry (i.e. $[Ca^{2+}]_o$ was reduced to 1 mM), and frequency of CTCs was greatly reduced at concentrations lower than 1 mM. Our concept is that $Ca^{2+}$ entry couples to CICR in ICC. Reducing the driving force for $Ca^{2+}$ entry would be expected to reduce the probability for effective coupling to CICR. CICR would be negligible when $Ca^{2+}$ entry falls below threshold levels. Concentrations of thapsigargin and CPA that blocked $Ca^{2+}$ transients quantitatively in other ICC

(*Drumm et al., 2020a*; *Drumm et al., 2019a*; *Drumm et al., 2017*) caused a partial block of $Ca^{2+}$ transients in ICC-SM. In fact, these drugs caused a marked narrowing of the duration of the CTCs, and this led us to analyze the temporal characteristics of $Ca^{2+}$ transients within clusters. $Ca^{2+}$ transients at the beginning of the CTCs were unaffected by ryanodine and xestospongin C, but transients toward the end of the clusters were blocked. These data suggest that the initial $Ca^{2+}$ transients may result primarily from $Ca^{2+}$ entry, and CTCs are sustained by $Ca^{2+}$ release. This concept is also supported by previous studies showing inhibition of slow waves by blocking $Ca^{2+}$ entry (L-type and T-type $Ca^{2+}$ channels) (*Drumm et al., 2017*).

We have searched for a preparation of pacemaker ICC that would allow us to investigate the underlying pacemaker activity. We have speculated that stochastic $Ca^{2+}$ release events, occurring in all ICC (*Sanders et al., 2014a*), are responsible for the spontaneous transient depolarizations (STDs) observed in patch clamp recordings from isolated ICC (*Zhu et al., 2009*; *Zhu et al., 2011*). No simultaneous recordings of $Ca^{2+}$ transients and membrane currents or potentials changes have been achieved yet, and the expected link between $Ca^{2+}$ transients and STDs is based on the fact that these events have common pharmacology and sensitivity to drugs that interfere with $Ca^{2+}$ release (*Zhu et al., 2015*). $Ca^{2+}$ transients and the spontaneous transient inward currents (STICs) due to activation of ANO1 channels result in spontaneous transient membrane depolarizations (STDs). Temporal summation of STDs is likely to be the generator potentials that activate T-type or L-type $Ca^{2+}$ currents and initiate propagating slow wave events. In this concept it is logical to suggest that inhibition of $Ca^{2+}$ release should reduce the duration of the CTCs, and inhibition of $Ca^{2+}$ entry should inhibit the organizing influence of $Ca^{2+}$ entry and block CTCs. When CTCs are blocked, stochastic $Ca^{2+}$ transients may be unleashed, as occur in ICC-IM (*Drumm et al., 2019a*) and ICC-DMP (*Baker et al., 2016*) that lack expression of voltage-dependent $Ca^{2+}$ entry mechanisms. Block of CTCs and unmasking of stochastic $Ca^{2+}$ transients was accomplished by hyperpolarization with pinacidil and reduction in the availability of L-type and T-type $Ca^{2+}$ channels with nicardipine and NNC 55–0396. ICC-SM, as imaged in the current experiments provide a potent model for investigating basic pacemaker mechanisms and what happens to these events in response to neurotransmission, hormonal and paracrine inputs and pathological or inflammatory conditions.

Previous studies have supported a role for store-operated $Ca^{2+}$ entry (SOCE) in maintaining $Ca^{2+}$ release events in ICC (*Zheng et al., 2018*; *Drumm et al., 2019a*). This is logical because $Ca^{2+}$ release is extremely dynamic in ICC, and it is likely that $Ca^{2+}$ stores would be depleted without an effective recovery mechanism. SOCE depends upon the expression of Orai channels and the ER delimited activator of Orai, STIM, that senses ER $Ca^{2+}$ and binds to and activates Orai channels when $Ca^{2+}$ depletion of the ER occurs (*Butorac et al., 2020*). STIM and Orai are expressed in colonic ICC (*Lee et al., 2017*). However, an antagonist of Orai, GSK7975A, reduced the frequency of CTCs, but failed to block these events at a concentration effective in blocking $Ca^{2+}$ transients in small intestinal ICC-MY (*Zheng et al., 2018*). STIM/Orai interactions appear to have a role in $Ca^{2+}$ store maintenance in ICC-SM, but our data suggest that $Ca^{2+}$ entry via L-type and T-type $Ca^{2+}$ channels also provide $Ca^{2+}$ entry mechanisms that may contribute to store refilling. It could also be suggested that GSK7975A, a well-known antagonist for Orai1 ($IC_{50}$ = 4 μM; *Derler et al., 2013*), is less effective on Orai2, the dominant isoform expressed in ICC-SM. However, studies on cortical neurons that express only Orai2 showed about 50% block of SOCE by GSK7975A (5 μM) (*Chauvet et al., 2016*).

In summary ICC-SM, as suggested from dissection and electrophysiological experiments (*Smith et al., 1987a*), are pacemaker cells distributed in an electrically coupled network along the submucosal surface of the CM layer. Our experiments demonstrate that $Ca^{2+}$ transients in ICC-SM couple to activation of $Ca^{2+}$ transients and contractions in neighboring SMCs. The contractile events, called 'ripples' by some authors in describing integrated colonic contractions (*Corsetti et al., 2019*; *D'Antona et al., 2001*), summate with the larger amplitude contractions emanating from the myenteric region of the *tunica muscularis* to produce mixing and propagated movements characteristic of colonic motility (*Rosli et al., 2020*). Data from this study suggest that voltage-dependent $Ca^{2+}$ entry serves at least four important functions in the pacemaker activity of ICC-SM: (i) Propagation of activity within the ICC-SM network depends upon voltage-dependent $Ca^{2+}$ entry, and the functions and voltage-dependent properties of three types of $Ca^{2+}$ conductances appear to provide a safety factor that tends to preserve pacemaker activity over a broad range of membrane potentials (see *Figure 15* and *Figure 15—figure supplement 1*). (ii) $Ca^{2+}$ entry is the mechanism that organizes $Ca^{2+}$ release events into CTCs. These events constitute the $Ca^{2+}$ waves that propagate through ICC-SM networks,

and cause slow wave depolarizations by activation of ANO1 channels. (iii) $Ca^{2+}$ entry also appears to contribute to refilling of stores, as pacemaker activity was not as immediately dependent upon SOCE as in other ICC (*Zheng et al., 2018*; *Drumm et al., 2019a*). (iv) The observations that treatments expected to reduce $Ca^{2+}$ release from stores and reduce coupling between $Ca^{2+}$ entry and CICR reduced, but did not block CTCs, may indicate that transient $Ca^{2+}$ entry (sparklets), possibly through activation of ANO1 channels and depolarization, may underlie the pacemaker functions of ICC-SM. Additional studies will be necessary to resolve these hypotheses in finer detail. The preparation of excised submucosal tissue with adherent ICC-SM removes movement artifacts from imaging and is likely to provide a powerful tool for improving resolution of pacemaker mechanisms and determining how regulatory and pathophysiological factors affect basic pacemaker mechanisms.

# Materials and methods

## Key resources table

| Reagent type (species) or resource | Designation | Source or reference | Identifiers | Additional information |
|---|---|---|---|---|
| Antibody | Anti- c-Kit (Goat polyclonal) | R and D Systems | Cat# AF1356, RRID:AB_354750 | IHC (1:500) |
| Antibody | Alexa-488 (donkey anti-goat IgG) | Invitrogen/ Thermo Fisher Scientific | (Cat# A32814, RRID:AB_2762838) | IHC (1:1000) |
| Sequence-based reagent | *Kit, Ano1, Myh11, Uchl1, Cacna1c Cacna1d, Cacna1g Cacna1h Orai1,Orai Orai3* | This paper | PCR primers | Suppl. Table 1 |
| Chemical compound, drug | NNC 55–0396, TTA-A2 | Alomone Labs | Cat# N-206 Cat# T-140 | |
| Chemical compound, drug | 2-APB, tetracaine, nicardipine, pinacidil, EGTA | Millipore-Sigma | Cat# D9754; Cat# T7508; Cat# N7510; Cat# P154; Cat# E4378 | |
| Chemical compound, drug | Thapsigargin, isradipine, Z-944, CPA, ryanodine | Tocris Bioscience | Cat# 1138/1; Cat# 2004/10; Cat# 6367/10; Cat# 1235/10; Cat# 1329/1 | |
| Chemical compound, drug | GSK 7975A | Aobious | Cat# AOB4124 | |
| Chemical compound, drug | Xestospongin C (XeC) | Cayman Chemical | Cat# 64950 | |
| Software, algorithm | STMapAuto, $Ca^{2+}$ Analysis Software | https://github.com/ gdelvalle99/STMapAuto | https://doi.org/10.1016/j.ceca.2020.102260 | |

## Animals

$Kit^{+/copGFP}$ mice (B6.129S7-$^{Kittm1Rosay/J}$; 5–8 wk old) were bred in house (*Ro et al., 2010*). GCaMP6f-floxed mice (Ai95 (RCL-GCaMP6f)-D) and C57BL/6 mice, their wild-type siblings, were purchased from Jackson Laboratories (Bar Harbor, MN, USA). Kit-iCre mice (c-Kit$^{+/Cre-ERT2}$) were gifted from Dr. Dieter Saur (Technical University Munich, Munich, Germany).

## Generation of Kit-iCre-GCaMP6f/Acta2-RCaMP1.07 mice

*Acta2-RCaMP1.07* mice (*tg(RP23-370F21-RCaMP1.07)B3-3Mik/J*) express the fluorescent $Ca^{2+}$ indicator RCaMP1.07 in SMCs under the control of the Acta2 locus promoter/enhancer regions were obtained from Jackson Laboratories (Bar Harbor, MN, USA). To generate cell-specific expression in two distinct cell types (ICC and SMCs) *Acta2-RCaMP1.07* mice were bred with $Kit^{Cre-ERT2}/$ $GCaMP6f^{fl/fl}$ mice. The offspring *Kit-iCre-GCaMP6f/Acta2-RCaMP1.07* mice were identified by genotyping after receiving tamoxifen which served to delete the STOP cassette in the Cre-expressing cells; resulting in the expression of the fluorescent $Ca^{2+}$ indicator protein, GCaMP6f. These mice allowed simultaneous, dual color imaging of ICC and SMCs. iCre mice were injected with tamoxifen (TAM; Intraperitoneal injection; IP) at 6–8 weeks of age (2 mg of TAM for three consecutive days), as

described previously (*Baker et al., 2016*), to induce activation of the Cre recombinase and expression of optogenetic sensors. Mice were used for experiments 10–15 days after the tamoxifen injections. On days of experiments the mice were anaesthetized by inhalation of isoflurane (Baxter, Deerfield, IL, USA) and killed by cervical dislocation before excision of tissues.

The animals used, protocols performed and procedures in this study were in accordance with the National Institutes of Health Guide for the Care and Use of Laboratory Animals and approved by the Institutional Animal Use and Care Committee at the University of Nevada, Reno (IACUC; Protocol: 00053).

## Tissue preparation

Colonic segments (2 cm in length, proximal region) were removed from mice after an abdominal incision and placed in Krebs-Ringer bicarbonate solution (KRB). The tissues were cut along the mesenteric border and intraluminal contents were washed away with KRB. Tissues were prepared by blunt dissection in two ways: (1) The submucosa layer was isolated after carefully removing the mucosal layer and the *tunica muscularis.* (2) The submucosa layer was left attached to the *tunica muscularis* after removal of the mucosa. The isolated submucosal layer preparation provided better imaging of ICC-SM by eliminating motion artifacts associated with muscle contractions. We used the isolated submucosal layer preparation in most cases in this study with the exception of two experiments where muscle attachments were necessary to test important questions (see Results).

## Immunohistochemistry

Colonic tissues from wild-type mice were processed to assess distribution of c-Kit immunoreactivity. Whole mounts of submucosal layer after removing the mucosa and *tunica muscularis* were fixed in 4% paraformaldehyde and visualized as described previously (*Sanders et al., 2014b*). Briefly, after block with 1% bovine serum albumin, colonic tissues were incubated with a polyclonal antibody raised against c-Kit (mSCFR, R and D Systems, MN, USA; 1:500 dilution in 0.5% Triton-X working solution) for 48 hr. Immunoreactivity was detected using Alexa-488 labeled donkey anti-goat IgG (1:1000 in PBS; Invitrogen, NY, USA). Colonic tissues were visualized using a Zeiss LSM 510 confocal microscope and images were constructed using Image J software (National Institutes of Health, MD, USA, http://rsbweb.nih.gov/ij). ICC (copKit) images were visualized using a spinning-disk confocal system (CSU-W1; spinning disk, Yokogawa Electric, Tokyo, Japan).

## Cell sorting and quantitative PCR

$Kit^{+/copGFP}$ mice (B6.129S7-$^{Kittm1Rosay/J}$; 5–8 wks old) were used for evaluations of gene expression in ICC-SM. Cell-specific expression of the fluorescent reporter allows unequivocal identification of ICC (*Ro et al., 2010*). After cell dispersion, ICC-SM were sorted by fluorescence-activated cell sorting (FACS) and evaluated for purity as previously described (*Baker et al., 2016*). Total RNA was isolated using an Illustra RNAspin Mini RNA Isolation Kit (GE Healthcare). qScript cDNA SuperMix (Quanta Biosciences), used according to the manufacturer's instructions, was used to synthesize first-strand cDNA. Quantitative PCR (qPCR) was performed using Fast Sybr Green chemistry on the 7900HT Fast Real-Time PCR System (Applied Biosystems) and gene-specific primers (*Supplementary file 1*). Regression analysis was performed to generate standard curves from the mean values of technical triplicate qPCRs of log10 diluted cDNA samples. Evaluation of gene expression in ICC-SM was compared with expression in the unsorted cells from the submucosal tissue of $Kit^{+/copGFP}$ mice.

## Ca²⁺ imaging

The isolated/intact submucosal layers were pinned to Sylgard coated dish and perfused with KRB solution at 37°C for a 60-min equilibration period. $Ca^{2+}$ imaging was performed using a spinning-disk confocal system (CSU-W1; spinning disk, Yokogawa Electric, Tokyo, Japan) mounted on an upright Nikon Eclipse FN1 microscope equipped with several water immersion Nikon CFI Fluor lenses (10 × 0.3 NA, 20 × 0.5 NA, 40 × 0.8 NA, 60 × 0.8 NA and 100 × 1.1 NA) (Nikon Instruments, New York, USA). The system is equipped with two solid-state laser lines of 488 nm and 561 nm. The laser lines are combined with a borealis system (ANDOR Technology, Belfast, UK) to increase laser intensity and uniformity throughout the imaging FOV. The system also has two high-speed electron multiplying charged coupled devices (EMCCD) cameras (Andor iXon-Ultra 897 EMCCD Cameras;

ANDOR Technology, Belfast, UK) to allow dual-color imaging simultaneously and maintain sensitive and fast speed acquisition at full frame of 512 × 512 active pixels as previously described (Baker et al., 2015). Briefly, images were captured, and image sequences were collected at 33 to 50 fps using MetaMorph software (MetaMorph Inc, TN, USA). In experiments with pharmacological agents, a control activity period of (30 s) was recorded prior of drug application into the chamber for 15 min.

### Ca²⁺ imaging analysis

Movies of $Ca^{2+}$ transients in ICC-SM were imported, preprocessed and analyzed using a combination of three image analysis programs: (1) custom build software (Volumetry G8d, Dr. Grant Hennig); (2) Fiji/Image J (National Institutes of Health, MD, USA, http://rsbweb.nih.gov/ij); (3) Automated Spatio Temporal Map analysis plugin (STMapAuto), https://github.com/gdelvalle99/STMapAuto as described previously (Drumm et al., 2017; Drumm et al., 2019b; Leigh et al., 2020). Briefly, movies of $Ca^{2+}$ transients (stacks of TIFF images) were imported into Volumetry G8d and motion stabilized, background subtracted and smoothed (Gaussian filter: 1.5 × 1.5 µm, StdDev 1.0). A particle analysis routine was employed using a flood-fill algorithm to enhance $Ca^{2+}$ transient detection. particles (PTCLs) representing the areas of active $Ca^{2+}$ signals in cells were saved as a coordinate-based particle movie and combined area and total number of PTCLs were calculated. To better isolate firing sites, only those particles that did not overlap with any particles in the previous frame but overlap with particles in the subsequent 70 ms were considered firing/initiation sites. To show the overall regions in cells where $Ca^{2+}$ transients occurred, PTCLs were summed throughout the video to create an image of their occurrence. The spatial information in the $Ca^{2+}$ occurrence maps data is an indication of each firing site that gives information on their temporal activation and provides no information on their spatial spread. This was implemented in our analysis to accommodate a large number of firing sites and effectively plot them in a 2D occurrence map.

### Drugs and solutions

All tissues were perfused continuously with KRB solution containing (mmol/L): NaCl, 5.9; $NaHCO_3$, 120.35; KCl, 1.2; $MgCl_2$, 15.5; $NaH_2PO_4$, 1.2; $CaCl_2$, 2.5; and glucose, 11.5. The KRB solution was warmed to a physiological temperature of 37 ± 0.3°C and bubbled with a mixture of 97% $O_2$ – 3% $CO_2$. For experiments utilizing external solutions with 0 $[Ca^{2+}]_o$, $CaCl_2$ was omitted and 0.5 mM ethylene glycol-bis (β-aminoethyl ether)-N, N, N', N'–tetraacetic acid (EGTA) was added to the solution. NNC 55–0396 and TTA-A2 were purchased from Alomone Labs (Jerusalem, Israel). 2-aminoethyldiphenylborinate (2-APB), tetracaine, nicardipine, pinacidil was purchased from Millipore-Sigma (St. Louis, Missouri, USA). Thapsigargin, isradipine, Z-944, CPA and ryanodine were purchased from Tocris Bioscience (Ellisville, Missouri, USA). GSK 7975A was purchased from Aobious (Aobious INC, MA, USA), and xestospongin C (XeC) was purchased from Cayman Chemical (Michigan, USA).

### Statistical analysis

Data is presented as the mean ± standard error unless otherwise stated. Statistical analysis was performed using either a Students $t$-test or one-way ANOVA with a Tukey post hoc test where appropriate. In all tests, $p < 0.05$ was considered significant. When describing data, $n$ refers to the number of animals used in a dataset. Probabilities < 0.05 are represented by a single asterisk (*), probabilities < 0.01 are represented by two asterisks (**), probabilities < 0.001 are represented by three asterisks (***) and probabilities < 0.0001 are represented by four asterisks (****). All statistical tests were performed using GraphPad Prism 8.0.1 (San Diego, CA).

## Acknowledgements

The authors extend their sincere appreciation to Yulia Bayguinov for assistance with immunohistochemical experiments, Lauren O'Kane for assistance with qPCR experiments and David White and Emily Fox for assistance with FACS.

## Additional information

### Funding

| Funder | Grant reference number | Author |
|---|---|---|
| National Institute of Diabetes and Digestive and Kidney Diseases | R01 DK-120759 | Salah A Baker<br>Kenton M Sanders |
| National Institute of Diabetes and Digestive and Kidney Diseases | R01 DK-078736 | Caroline A Cobine |

The funders had no role in study design, data collection and interpretation, or the decision to submit the work for publication.

### Author contributions

Salah A Baker, Conceptualization, Resources, Data curation, Software, Formal analysis, Supervision, Funding acquisition, Validation, Investigation, Visualization, Methodology, Writing - original draft, Project administration, Writing - review and editing; Wesley A Leigh, Data curation, Formal analysis, Investigation, Methodology; Guillermo Del Valle, Formal analysis; Inigo F De Yturriaga, Data curation, Formal analysis, Investigation; Sean M Ward, Validation, Writing - review and editing; Caroline A Cobine, Data curation, Formal analysis, Methodology, Writing - review and editing; Bernard T Drumm, Formal analysis, Investigation, Methodology, Writing - review and editing; Kenton M Sanders, Conceptualization, Resources, Data curation, Supervision, Funding acquisition, Validation, Writing - original draft, Project administration, Writing - review and editing

### Author ORCIDs

Salah A Baker (iD) https://orcid.org/0000-0002-1514-6876

### Ethics

Animal experimentation: The animals used, protocols performed and procedures in this study were in accordance with the National Institutes of Health Guide for the Care and Use of Laboratory Animals and approved by the Institutional Animal Use and Care Committee at the University of Nevada, Reno (IACUC; Protocol# 00053).

### Decision letter and Author response

Decision letter https://doi.org/10.7554/eLife.64099.sa1
Author response https://doi.org/10.7554/eLife.64099.sa2

## Additional files

### Supplementary files

• Supplementary file 1. Table summary of gene primers sequences of *Kit, Ano1, Myh11, Uchl1, Cacna1c, Cacna1d, Cacna1g,* Cacna1h, *and Orai1-3.*

• Transparent reporting form

### Data availability

All data generated or analysed during this study are included in the manuscript and supporting files.

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
