## [Decision Letter]

**Acceptance summary:**

Gastrointestinal motility is controlled by the activity of interstitial cells of Cajal (ICC). Here the authors elucidate the calcium signals that drive the activity of submucosal ICCs which in turn elicit phasic contractions of the colon. The authors demonstrate that the calcium signals depend on calcium entry into submucosal ICCs.

**Decision letter after peer review:**

Thank you for submitting your article "Ca^2+^ signaling driving pacemaker activity in submucosal interstitial cells of Cajal in the colon" for consideration by *eLife*. Your article has been reviewed by three peer reviewers, one of whom is a member of our Board of Reviewing Editors, and the evaluation has been overseen by Richard Aldrich as the Senior Editor. The following individual involved in review of your submission has agreed to reveal their identity: Michael J Davis (Reviewer #3).

The reviewers have discussed the reviews with one another and the Reviewing Editor has drafted this decision to help you prepare a revised submission.

Summary:

In the manuscript entitled "Ca^2+^ signaling driving pacemaker activity in submucosal interstitial cells of Cajal in the colon", the authors investigated the role of different Ca^2+^ pathways on pacemaker (slow wave) activity in the mouse colon. While much of the investigation uses techniques and tools used to previously by the authors study Ca^2+^ dynamics in ICC in other regions of the gut (small intestine), they have refined the colonic tissue preparation to remove motion artifacts and have used dual Ca^2+^ indicator mice to better understand how ICC-SM influence smooth muscle Ca^2+^ dynamics and contraction. Overall, this solid study has used a comprehensive battery of drugs that interfere with Ca^2+^ signaling to demonstrate that extracellular Ca^2+^ entry is a particularly important component of pacemaker activity in ICC-SM.

Revisions:

1) Paradoxically, firing failure usually causes subsequent pacemaker events to release more Ca^2+^ (amplitude/duration), and are likely to result in stronger/more complete activation of the SM. In terms of motility, it is possible (and likely) that "stronger" pacing at half control frequency may impart more useful force to contents and potentially improve motility. As the authors are well aware, this may be the default situation in the stomach (2pacemaker:1contraction). An analysis of a number of common parameters (amplitude, duration, upstroke velocity) performed on each observable event (rather than on all the events in the recording period) would be important to better assess the effects of the drugs used.

2) Interestingly, the authors show a different aspect of the "recovery" effect on the coordination of intracellular release site firing – which is somewhat unexpected, and important. Decreasing extracellular Ca^2+^ began to cause instabilities in the coordination/synchronization in site firing at 1mM while maintaining a frequency close to that observed at 2.5mM. At 0.5mM, firing failure occurred, but in the examples where firing occurred, it was the most synchronous / coordinated observed in the entire experiment. This, again, could indicate that lengthening the duration of "recovery" after a pacemaker event is one of the major factors determining the coordination of both intracellular (site) firing, and by extension, intracellular (network) activity. In contrast to the author's conclusions, this process does not appear to be that dependent on extracellular Ca^2+^ entry and needs to be commented on. In addition, lowering calcium will also affect membrane surface charge, which in itself will affect the electrical field experience by voltage-dependent ion channels.

3) The spatial segmentation of dynamic Ca^2+^ data into PTCLs based on "first" event seems arbitrary, as the degree to which Ca^2+^ spreads from the position of the "first" event appears to be quite variable. The image illustrating individual sites appears to be hand-drawn. If site demarcation was somewhat arbitrary, this should be clarified.

4) The cell density value reported (95 cells per mm2) seems low by an order of magnitude. If the quoted average minimum separation distance is 30µm, one would expect 33 cells across a 1mm in 1D. Using the same separation distance, in 2D, one would expect 33 cells x 33 cells = ~1000 cells per square mm.

5) While a correlation exists between the ICC-SM Ca^2+^ events, smooth muscle Ca^2+^ events and resultant micromotion (Figure 2) in general terms, the traces shown in Figure 2E do not show a 1:1 correlation of the event waveforms. While some variability is expected between micromotions and fluorescence signals due to unknown dynamic/static stress and strain in the gut wall, one would expect the Ca^2+^ event waveforms in ICC-SM and SM to be more correlated given the tight electrical coupling via gap junctions. While all the presented Ca^2+^ events are biphasic in nature, the first event ICC-SM shows a clear separation between the 2 peaks. The time between the 2 peaks appears sufficient to allow for re-stimulation the SM (~1s), given the fluorescence has returned 80-90% back to baseline – yet there is no inflection or change in slope in the SM Ca^2+^ event. The authors should comment on this disparity and qualify under which conditions (frequency/amplitude/? range) the correlation holds true using cross-correlation analysis. Differentiation and distortion vectors would also better locate contraction epicenters in videos and differentiation of traces would provide a more reliable landmark (max dI/dT) to compare time delays.

6) The stomach and small intestine generate slow waves in the outermost ICC layer (myenteric layer: ICC-MY) that co-activates circular and longitudinal muscle layers. Activation of colonic CM from submucosal ICC layer prevents slow waves from directly co-activating longitudinal SM, and this is evident in more independent and varied longitudinal motions (tenia coli tone, greater neural control). The role of colonic ICC-MY is more confusing but appears to synchronize activity in certain conditions, which has been proposed to change the excitability of CM and possibly other layers in the gut wall. As such, without knowing the output from colonic ICC-MY, attempts to correlate CM activity to ICC-SM activity may be highly variable and unreliable. A comment on this would be helpful.

7) The conclusion that Ca^2+^ entry through L-type Ca^2+^ channels is vital to the generation of pacemaker activity is valid with regard to network activity, but the effect on intracellular release sites seems less clear. In Figure 9 A, there are 8-9 well defined "hot spots" demarcated by their white/red color (denoting how long the site was active). The cytoplasm surrounding these hotspots shows a lower occurrence of activity, consistent with an occasional propagating intracellular Ca^2+^ wave. Interestingly, all of the main hotspots remained after nicardipine, albeit firing at a much lower frequency. Wouldn't this suggest that release sites are not intrinsically dependent on L-type Ca^2+^ entry and that the L-Type channels are more involved in facilitating spread of Ca^2+^ through the cytoplasm?

8) Have the authors used an L-type Ca^2+^ channel opener (BayK8644) on the ICC-SM preparation? This would yield important information regarding the role of channel inactivation/"recovery" and the importance of entrainment and limiting ectopic activity on the formation of coherent pacemaker activity.

9) Wouldn't large effect of NNC after nicardipine (Figure 12) suggest that T-type Ca^2+^ channel entry is equally important is L-type entry. The frequency of control activity prior to the addition of NNC (Figure 10) was quite high and is likely to have predisposed cells to firing failure to a much larger degree as slower control frequencies.

10) Raw data ± SE was presented in the both the text and in graphical form, particularly when presenting % data. The text in the Results section or the latter parts of many figures could be reduced considerably and easier to read/ view without the repetition. This would considerably raise the impact of the work.

11) The results of Pinacidil alone (Figure 11) could easily be removed as they didn't reach significance; at the very least, moved to the supplementary section.

12) The effects of pinacidil on Ca channel blockers could perhaps be best illustrated by plotting the red and blue bars in Figure 11 with the normalized results in Figures 9 and 10 to illustrate any effect. “Tendency” statements that should be removed include: “Pinacidil showed a tendency toward increased Ca^2+^ transient firing, but this effect did not reach statistical significance” and “Reduction in the duration of Ca^2+^ transients was associated with a tendency for an increase in frequency, but this parameter did not reach significance”

13) The Xestospongin modulation of the Ca store signalling in ICC-SM is not significant and probably doesn't require illustration or it should be moved to the supplementary data section.

14) The qPCR analyses provide evidence for message for those two channels and the use of three different pharmacological inhibitors (for Cav3) with similar effects (Figure 10, Figure 10—figure supplement 2) support their functional expression. However, can the authors rule out the possibility that NNC 55-0396, TTA-A2 and Z-944 are producing their respective effects on calcium transients by partial inhibition of L-type channels? e.g., does a lower concentration of nicardipine produce similar effects to those agents?

15) Pinacidil was used to activate KATP channels in the muscle layer, when it remained connected to the ICC-SM layer; this presumably(?) led to hyperpolarization of the ICC layer due to electrical coupling between the layers. Is that correct? (If so that assumption was not clearly stated). The differential effects of the calcium channel inhibitors before and after PIN were then used as evidence that voltage-activated calcium channels with more negative thresholds played a role when PIN was used. But shouldn't the effects of isradipine also have been tested in the presence of PIN in order to make conclusions about the contribution of Cav1.3 channels? Supporting electrophysiology data showing the extent of PIN-induced hyperpolarization in ICC-SM cells would aid in interpretation of these data.

16) The authors state that ICC in general do not express KATP channels, but wasn't that analysis made for another type of ICC (ICC-MY?)? If that is so, they should check for Kir6 and SUR expression in these purified ICC-SM cells.

17) Previous studies of ICC-SM in mouse have used 0.01 – 1 μm nifedipine (Yoneda et al., 2002; 2003) and Ni2+ to dissect out contributions of L-and T- type channels to the initial and plateau phases of the slow waves in these cells. That data should be discussed at least to some extent. Do the authors think that the pattern of calcium events (fast and transient vs prolonged) correspond to the initial and sustained phases of the slow waves measured in previous studies, in which it was concluded that T- and L-type calcium channels respectively contributed?

---

## [Author Response]

Revisions:1) Paradoxically, firing failure usually causes subsequent pacemaker events to release more Ca^2+^ (amplitude/duration), and are likely to result in stronger/more complete activation of the SM. In terms of motility, it is possible (and likely) that "stronger" pacing at half control frequency may impart more useful force to contents and potentially improve motility. As the authors are well aware, this may be the default situation in the stomach (2pacemaker:1contraction). An analysis of a number of common parameters (amplitude, duration, upstroke velocity) performed on each observable event (rather than on all the events in the recording period) would be important to better assess the effects of the drugs used.

We analyzed the Ca^2+^ firing sites in ICC-SM using particle analysis method. This analysis is based on temporal characteristics to define firing sites. The number of Ca^2+^ firing events is enormous and variable cycle-to-cycle, as firing sites can fire multiple times. The particle analysis method relies on binary transformation to effectively define and segment Ca^2+^ firing sites. Therefore, the intensity amplitude of each firing site is not available, but this binary transformation enables us to effectively quantify the large number of sites within the FOV in terms of temporal information and provides parameters of Ca^2+^ event size, count, duration and frequency.

Following the reviewer's suggestion, in addition to firing sites size and frequency count, we analyzed all data sets under control and drug conditions and added the duration information for Ca^2+^ particle events to aid the readers to better understand the observed drug effects. The event duration data are included in updated figures (Figures 7, 9, 10, 12, 13 and 14) and in the Results section.

2) Interestingly, the authors show a different aspect of the "recovery" effect on the coordination of intracellular release site firing – which is somewhat unexpected, and important. Decreasing extracellular Ca^2+^ began to cause instabilities in the coordination/synchronization in site firing at 1mM while maintaining a frequency close to that observed at 2.5mM. At 0.5mM, firing failure occurred, but in the examples where firing occurred, it was the most synchronous / coordinated observed in the entire experiment. This, again, could indicate that lengthening the duration of "recovery" after a pacemaker event is one of the major factors determining the coordination of both intracellular (site) firing, and by extension, intracellular (network) activity. In contrast to the author's conclusions, this process does not appear to be that dependent on extracellular Ca^2+^ entry and needs to be commented on. In addition, lowering calcium will also affect membrane surface charge, which in itself will affect the electrical field experience by voltage-dependent ion channels.

We agree with the reviewer that reducing [Ca^2+^] _o_ to 0.5 mM caused a firing failure, which is most likely due to a reduction in Ca^2+^ entry and effectively maintaining store refilling and CICR mechanisms. This observation confirms the role of Ca^2+^ entry in maintaining Ca^2+^ release mechanisms as indicated by the reduction of the number of Ca^2+^ firing sites (Figure 7 K and L)

Following the reviewer’s suggestion, we analyzed the duration of Ca^2+^ firing sites under reduced [Ca^2+^] _o_ conditions (New Figure 7). There was no significant change in the average individual firing sites duration except when we reduce [Ca^2+^] _o_ to 1 mM, we noticed a significant increase. This observed increase in Ca^2+^ firing duration at 50% [Ca^2+^] _o_ may be providing a compensatory mechanism for pacemaker firing failure. Reducing [Ca^2+^] _o_ further had no significant change in individual firing site duration but the duration of the whole Ca^2+^ transient cycle (CTC) was reduced, and this is in line with previous paper (Ward et al. J Physiology 1992; https://doi.org/10.1113/jphysiol.1992.sp019303) that showed reducing [Ca^2+^] _o_ to 0.5 mM and 0.25 mM caused a firing failure in colonic slow waves that was accompanied with a reduction in electrical slow wave duration in colonic tissues.

We agree with the reviewer that membrane surface charge could influence some experiments where the divalent ion concentration decreased as [Ca^2+^] _o_ was reduced. We have addressed this point in a previous paper with a series of cation replacement of Ca^2+^ by Mg^2+^ were tested on colonic electrical slow waves and found no significant differences in the responses (Ward et al., J Physiology 1992).

3) The spatial segmentation of dynamic Ca^2+^ data into PTCLs based on "first" event seems arbitrary, as the degree to which Ca^2+^ spreads from the position of the "first" event appears to be quite variable. The image illustrating individual sites appears to be hand-drawn. If site demarcation was somewhat arbitrary, this should be clarified.

The Ca^2+^ particle analysis used in the study relies on the calculation of the combined area and total number of PTCLs for the entire video. Flags (8-bit, bit-hashed) allowed entire PTCLs to be distinguished based on whether they met certain spatiotemporal (ST) conditions such as size and temporal characteristics. To show the overall regions in cells where Ca^2+^ transients occurred, PTCLs were summed throughout the video to create an image of their occurrence.

Yes. The spatial information in the Ca^2+^ occurrence maps data is an indication of each firing site that gives information on their temporal activation and provides no information on their spatial spread. This was implemented in our analysis to accommodate a large number of firing sites and effectively plot them in a 2D occurrence map. We clarified this point in the Materials and methods section.

4) The cell density value reported (95 cells per mm2) seems low by an order of magnitude. If the quoted average minimum separation distance is 30µm, one would expect 33 cells across a 1mm in 1D. Using the same separation distance, in 2D, one would expect 33 cells x 33 cells = ~1000 cells per square mm.

We thank the reviewer for raising this point. The original minimum separation distances between ICC-SM cells were calculated only with the inclusion of very closely associated ICC-SM cells. The ICC-SM network seems to have multiple processes that can vary in length, as visualized in Figure 1. Therefore, we reanalyzed our images and increased the number of images to provide more accurate cell density measurements. New cell density data was added to the Results section.

5) While a correlation exists between the ICC-SM Ca^2+^ events, smooth muscle Ca^2+^ events and resultant micromotion (Figure 2) in general terms, the traces shown in Figure 2E do not show a 1:1 correlation of the event waveforms. While some variability is expected between micromotions and fluorescence signals due to unknown dynamic/static stress and strain in the gut wall, one would expect the Ca^2+^ event waveforms in ICC-SM and SM to be more correlated given the tight electrical coupling via gap junctions. While all the presented Ca^2+^ events are biphasic in nature, the first event ICC-SM shows a clear separation between the 2 peaks. The time between the 2 peaks appears sufficient to allow for re-stimulation the SM (~1s), given the fluorescence has returned 80-90% back to baseline – yet there is no inflection or change in slope in the SM Ca^2+^ event. The authors should comment on this disparity and qualify under which conditions (frequency/amplitude/? range) the correlation holds true using cross-correlation analysis. Differentiation and distortion vectors would also better locate contraction epicenters in videos and differentiation of traces would provide a more reliable landmark (max dI/dT) to compare time delays.

We agree with the reviewer that the ICC network is coupled to SMCs via gap junctions providing an electric signal from ICC to neighboring SMCs. Because of the syncytial nature of the 2 networks, multiple ICC-SM can influence the activity of any given SMC. Thus, the overall signal in adjacent SMC represents a summation of inputs from many ICC-SM, and this may explain the observation of the reviewer.

In addition, ICC demonstrate complex patterns of Ca^2+^ firing in comparison to SMCs. Ca^2+^ transients in ICC activate ANO1 currents. Conduction of currents and depolarization increases the open probability of L-type Ca^2+^ channels in SMCs, resulting in the Ca^2+^ transients in these cells. The sequence of signals and coupling between Ca^2+^ transients and electrical responses is different in ICC and SMCs. In fact it is opposite: a rise in [Ca^2+^]_o_ initiates ANO1 currents and depolarization in ICC, and depolarization initiates openings of Ca^2+^ channels and Ca^2+^ entry in SMCs. We don’t anticipate that subtle changes in Ca^2+^ transients in one or a few ICC during a single cycle would have great influence on the Ca^2+^ signals in SMCs, as depolarizing currents from ICC activate Ca^2+^ action potentials. Thus, activation of SMCs may be independent of the exact waveforms of Ca^2+^ transients in ICC as long as threshold is reached. Another factor separating 1:1 waveform behavior in ICC and SMCs is the refractory properties of Ca^2+^ action potentials in SMC and Ca^2+^ in ICC are fundamentally different, causing differences in the temporal and overall “shaping” of responses. To clarify this point, we added comment on the contribution of multiple ICC to the overall SMC signals in the Discussion section.

6) The stomach and small intestine generate slow waves in the outermost ICC layer (myenteric layer: ICC-MY) that co-activates circular and longitudinal muscle layers. Activation of colonic CM from submucosal ICC layer prevents slow waves from directly co-activating longitudinal SM, and this is evident in more independent and varied longitudinal motions (tenia coli tone, greater neural control). The role of colonic ICC-MY is more confusing but appears to synchronize activity in certain conditions, which has been proposed to change the excitability of CM and possibly other layers in the gut wall. As such, without knowing the output from colonic ICC-MY, attempts to correlate CM activity to ICC-SM activity may be highly variable and unreliable. A comment on this would be helpful.

We agree with the reviewer that several types of ICC exist in the colon, and all, through coupling of Ca^2+^ transients and activation of ANO1 channels, can contribute to colonic excitability. Both ICC-SM and ICC-MY provide pacemaker activities in the colon. To understand the complex behaviors of these cells, a reductionist approach is first needed. In this study we focused on the intrinsic pacemaker role of ICC-SM in driving adjacent smooth muscle. Most of the experiments in the study, used an isolated ICC-SM preparation to define Ca^2+^ signals in ICC-SM with minimal motion artifacts. To examine if ICC-SM drive activity in CM, we used a preparation that included attached SMCs. It is true these preparations included additional types of ICC that might also influence the ultimate behaviors of SMCs. Therefore, we cannot exclude the fact that the signals in SMCs under the conditions of these specific experiments are influenced by additional types of ICC. We know from this study and previously published papers that ICC-SM in the mouse colon generates slow waves at ~ 15 /min whereas ICC-MY ~2/min. Contractions observed in our experiments followed the frequency of behaviors in ICC-SM. Possible influences from additional cells within the muscle tissues were not evaluated in this study. To fully understand the highly integrated functions in colonic motility that result from the behaviors of many types of cells: ICC, SMCs and interstitial cells labeled with antibodies to PDGFRα (i.e. the SIP syncytium), enteric neural inputs and tuning of neural activity by glial cells, will require considerably more complex imaging approaches and refined techniques to limit movements without interfering with the electrical behaviors and Ca^2+^ dynamics of the cells involved.

At this point, we do not know if the activity of ICC-SM propagates through the CM to influence the behaviors of ICC-MY and longitudinal muscle or vice versa. Previous studies in larger animals have shown that pacemaker propagation declines as a function of distance from the submucosal region towards the myenteric region. Smith et al., 1987 AJP.

7) The conclusion that Ca^2+^ entry through L-type Ca^2+^ channels is vital to the generation of pacemaker activity is valid with regard to network activity, but the effect on intracellular release sites seems less clear. In Figure 9 A, there are 8-9 well defined "hot spots" demarcated by their white/red color (denoting how long the site was active). The cytoplasm surrounding these hotspots shows a lower occurrence of activity, consistent with an occasional propagating intracellular Ca^2+^ wave. Interestingly, all of the main hotspots remained after nicardipine, albeit firing at a much lower frequency. Wouldn't this suggest that release sites are not intrinsically dependent on L-type Ca^2+^ entry and that the L-Type channels are more involved in facilitating spread of Ca^2+^ through the cytoplasm?

After the addition of nicardipine a few remnant Ca^2+^ release sites continued to fire throughout experiments (hot-spots). But after nicardipine these sites fired randomly, and their frequency was very low. Thus, we interpret these observations to mean that Ca^2+^ release is an intrinsic property of ICC, and it occurs from specific sites within cells. Ca^2+^ influx enhances the excitability of Ca^2+^ sites and is crucial for organizing Ca^2+^ release into clusters that cause sustained activation of ANO1 channels and electrical depolarizations (slow waves). Ca^2+^ entry organizes Ca^2+^ release via CICR. Ca^2+^ entry through L-type Ca^2+^ channels may also contribute to refilling of stores.

8) Have the authors used an L-type Ca^2+^ channel opener (BayK8644) on the ICC-SM preparation? This would yield important information regarding the role of channel inactivation/"recovery" and the importance of entrainment and limiting ectopic activity on the formation of coherent pacemaker activity.

We have used the L-type channel antagonist (Nicardipine) in this study to evaluate contributions of these channels, however we have not used an L-type channel agonist, such as BayK8644. It is an interesting suggestion, and we will utilize this compound in future studies.

9) Wouldn't large effect of NNC after nicardipine (Figure 12) suggest that T-type Ca^2+^ channel entry is equally important is L-type entry. The frequency of control activity prior to the addition of NNC (Figure 10) was quite high and is likely to have predisposed cells to firing failure to a much larger degree as slower control frequencies.

We agree with the reviewer that both L-type and T-type Ca^2+^ channels are present and may contribute to pacemaker activity in ICC-SM. However, the resting membrane potential of colonic muscles is usually not at highly negative potentials (e.g. ~45 mV; Koh et al. Neurogastroenterol Motil. 2012: doi: 10.1111/j.1365-2982.2012.01956.x). Thus, the relative importance of T-type Ca^2+^ channels may be reduced due to tonic inactivation of these channels. In cases of depolarized cells, L-type Ca^2+^ channels may dominate, as appears to happen under resting conditions in our experiments. T-type Ca^2+^ channels, however, can become more important if membrane potential is hyperpolarized, as indicated by our experiments when pinacidil was used to hyperpolarize cells. This shift in importance between the two conductances could be important in pathophysiological conditions associated with changes in the membrane potential. Having two conductance expressed with different voltage-dependencies provides a safety factor for maintenance of pacemaker activity, an important and novel observation of the present study.

10) Raw data ± SE was presented in the both the text and in graphical form, particularly when presenting % data. The text in the Results section or the latter parts of many figures could be reduced considerably and easier to read/ view without the repetition. This would considerably raise the impact of the work.

We thank the reviewer for this suggestion, we feel that including the Raw data ± SE in the Results section would provide the reader of details on the degree of drug effects and help in tabulate the results presented.

11) The results of Pinacidil alone (Figure 11) could easily be removed as they didn't reach significance; at the very least, moved to the supplementary section.

Following the reviewer's suggestion, we removed the effects of pinacidil alone to a new supplementary figure (Figure 11—figure supplement 1).

12) The effects of pinacidil on Ca channel blockers could perhaps be best illustrated by plotting the red and blue bars in Figure 11 with the normalized results in Figures 9 and 10 to illustrate any effect. “Tendency” statements that should be removed include: “Pinacidil showed a tendency toward increased Ca^2+^ transient firing, but this effect did not reach statistical significance” and “Reduction in the duration of Ca^2+^ transients was associated with a tendency for an increase in frequency, but this parameter did not reach significance”

Following the reviewer's suggestion, we removed the effects of pinacidil alone to a new supplementary figure. We also removed all tendency statements from the Results section.

13) The Xestospongin modulation of the Ca store signalling in ICC-SM is not significant and probably doesn't require illustration or it should be moved to the supplementary data section.

Following the reviewer's suggestion, we removed the effects of Xestospongin into a new supplementary figure (Figure 13—figure supplement 1).

14) The qPCR analyses provide evidence for message for those two channels and the use of three different pharmacological inhibitors (for Cav3) with similar effects (Figure 10, Figure 10—figure supplement 1) support their functional expression. However, can the authors rule out the possibility that NNC 55-0396, TTA-A2 and Z-944 are producing their respective effects on calcium transients by partial inhibition of L-type channels? e.g., does a lower concentration of nicardipine produce similar effects to those agents?

A previous report has examined the selectivity of these drugs. Ming li et al. 2005 (DOI: 10.1111/j.1527-3466.2005.tb00164.x) demonstrated that 100 microM NNC 55-0396 has no detectable effect on high voltage-activated currents and does not produce metabolite that causes inhibition of L-type Ca^2+^ channel.

15) Pinacidil was used to activate KATP channels in the muscle layer, when it remained connected to the ICC-SM layer; this presumably(?) led to hyperpolarization of the ICC layer due to electrical coupling between the layers. Is that correct? (If so that assumption was not clearly stated). The differential effects of the calcium channel inhibitors before and after PIN were then used as evidence that voltage-activated calcium channels with more negative thresholds played a role when PIN was used. But shouldn't the effects of isradipine also have been tested in the presence of PIN in order to make conclusions about the contribution of Cav1.3 channels? Supporting electrophysiology data showing the extent of PIN-induced hyperpolarization in ICC-SM cells would aid in interpretation of these data.

In the current study, pinacidil application was used to produce a membrane hyperpolarization in SMCs via activation of KATP channels and ICC as they are electrically coupled. We added a statement in the Results section to clarify the rationale of pinacidil application.

We agree with the reviewer point that these experiments do not directly incorporate potential involvement of Cav 1.3 channels. Data in this study show that ICC-SM Ca^2+^ transients depend upon Ca^2+^ influx via L-type Ca^2+^ channels, and since isradipine had nearly the same or a lesser effect on Ca^2+^ transients than nicardipine, Ca_V_ 1.2 appear dominant as the Ca^2+^ entry mechanism. This set of experiments were targeted to determine the contributions of T-type Ca^2+^ channels. The results demonstrate that at negative potentials, where the open probability of L-type Ca^2+^ channels is low, T-type Ca^2+^ channels are available to facilitate Ca^2+^ influx and sustain pacemaker activity. We agree that patch clamp experiments would further support the conclusions, and we will incorporate them in future studies.

16) The authors state that ICC in general do not express KATP channels, but wasn't that analysis made for another type of ICC (ICC-MY?)? If that is so, they should check for Kir6 and SUR expression in these purified ICC-SM cells.

ICC express message for Kir6 and SUR, however KATP currents cannot be resolved in colonic ICC (Huang et al. 2018). At present it is unknown whether Kir6 and SUR are not translated or not trafficked to the plasma membranes to create functional KATP channels in ICC.

17) Previous studies of ICC-SM in mouse have used 0.01 – 1 μm nifedipine (Yoneda et al., 2002; 2003) and Ni2+ to dissect out contributions of L-and T- type channels to the initial and plateau phases of the slow waves in these cells. That data should be discussed at least to some extent. Do the authors think that the pattern of calcium events (fast and transient vs prolonged) correspond to the initial and sustained phases of the slow waves measured in previous studies, in which it was concluded that T- and L-type calcium channels respectively contributed?

Both of Yoneda et al. papers (Yoneda et al., 2002; 2003) concluded that; L-type and T-type Ca^2+^ channels contribute to slow waves in the colon. L-type Ca^2+^ channels completely abolished slow waves where T-type channels are involved in the initial component of the slow wave. Our study highlighting the Ca^2+^ signaling in ICC-SM is consistent with conclusions of the Yoneda et al. papers. Although the comparison between the electrical activity and Ca^2+^ dynamics from different studies is difficult to clearly interpret. Remember that the electrical activity was recorded (most likely) from SMCs and we are imaging Ca^2+^ dynamics in most experiments exclusively from ICC. Figure 15 summarizes the sequence of events in ICC, and these events drive the depolarization-dependent activation of Ca^2+^ action potentials/ Ca^2+^ entry in SMCs. We have taken the reviewer’s suggestion and added more information about the conclusions of Yoneda et al. in the Discussion.